# QCircuitNet: A Large-Scale Hierarchical Dataset for Quantum Algorithm Design

## Abstract

Quantum computing is an emerging field recognized for the significant speedup it offers over classical computing through quantum algorithms. However, designing and implementing quantum algorithms pose challenges due to the complex nature of quantum mechanics and the necessity for precise control over quantum states. To address these challenges, we leverage AI to simplify and enhance the process. Despite the significant advancements in AI, there has been a lack of datasets specifically tailored for this purpose. In this work, we introduce QCircuitNet, a benchmark and test dataset designed to evaluate AI's capability in designing and implementing quantum algorithms in the form of quantum circuit codes. Unlike traditional AI code writing, this task is fundamentally different and significantly more complicated due to the highly flexible design space and the extreme demands for intricate manipulation of qubits. Our key contributions include: 1. The first comprehensive, structured universal quantum algorithm dataset. 2. A framework which formulates the task of quantum algorithm design for Large Language Models (LLMs), providing guidelines for expansion and potential evolution into a training dataset. 3. Automatic validation and verification functions, allowing for scalable and efficient evaluation methodologies. 4. A fair and stable benchmark that avoids data contamination, a particularly critical issue in quantum computing datasets. Our work aims to bridge the gap in available resources for AI-driven quantum algorithm design, offering a robust and scalable method for evaluating and improving AI models in this field. As we expand the dataset to include more algorithms and explore novel fine-tuning methods, we hope it will significantly contribute to both quantum algorithm design and implementation.

## 1 Introduction

Quantum computing is an emerging field in recent decades, which can attribute to the fact that algorithms on quantum computers may solve problems significantly faster than their classical counterparts. From the perspective of theoretical computer science, the design quantum algorithms have been investigated in various research directions - see the survey [Dalzell et al., 2023] and the quantum algorithm zoo [Zoo, 2024]. However, the design of quantum algorithms on quantum computers has been completed manually by researchers. This process is notably challenging due to highly flexible design space and extreme demands for a comprehensive understanding of mathematical tools and quantum properties.

Submitted to the 38th Conference on Neural Information Processing Systems (NeurIPS 2024) Track on Datasets and Benchmarks. Do not distribute.

For these reasons, quantum computing is often considered to have high professional barriers. As the discipline evolves, we aim to explore more possibilities for algorithm design and implementation in the quantum setting. This is aligned with recent advances among "AI for Science", including AlphaFold [Jumper et al., 2021], AlphaGeometry [Trinh et al., 2024], etc. Recently, large language models (LLMs) has also become crucial among AI for science approaches [Yang et al., 2024, Zhang et al., 2024, Yu et al., 2024]. Therefore, we attempt to gear LLMs for quantum algorithm design. As far we know, there has not been any dataset for AI in quantum algorithm design. Existing work combining quantum computing and AI are mostly targeting at exploiting quantum computing for AI; there are some papers that apply AI for quantum computing, but they consider niche problems [Nakayama et al., 2023, Schatzki et al., 2021] or limited functions [Tang et al., 2023, Fürrutter et al., 2024], not quantum algorithm datasets of general interest. See more discussions in Section 2.

**Key contributions.** In this work, we propose QCircuitNet, the first comprehensive, structured dataset for quantum algorithm design. Technically, QCircuitNet has the following key contributions:

- It formulates the task of quantum algorithm design for Large Language Models (LLMs), providing guidelines for expansion that may evolve to be a training dataset.
- It has automatic validation and verification functions, allowing for scalable and efficient evaluation.
- It provides a fair and stable benchmark that avoids data contamination, a particularly critical issue in quantum computing datasets.

## 2 Related Work

To the best of our knowledge, QCircuitNet is the first dataset tailored specifically for quantum algorithm design. Previous efforts combining quantum computing with artificial intelligence primarily fall under the category of Quantum Machine Learning (QML), which aims at leveraging the unique properties of quantum systems to enhance machine learning algorithms and achieve potential improvements over their classical counterparts [Schuld et al., 2015, Biamonte et al., 2017, Ciliberto et al., 2018]. Corresponding datasets often focus on encoding classical data into quantum states, which we may call "Quantum for AI". For instance, MNISQ [Placidi et al., 2023] is a dataset of quantum circuits representing the original MNIST dataset [LeCun et al., 1998] generated by the AQCE algorithm [Shirakawa et al., 2021]. Considering the intrinsic nature of quantum properties, another category of datasets focuses on collecting quantum data to demonstrate quantum advantages since classical machine learning methods could fail to characterize particular patterns of quantum data. For example, Nakayama et al. [2023] created a VQE-generated quantum circuit dataset for classification of variational ansatzes and shows the quantum supremacy on this task. NTangled [Schatzki et al., 2021] further emphasized on the different types and amounts of entanglement and composed quantum states with various multipartite entanglement for classification. While these datasets successfully demonstrate the supremacy of quantum computing, they address rather niche problems which might not have practical applications.

There have also been efforts in the direction of "AI for Quantum", which explores the possibility of leveraging the huge potential of AI to facilitate the advancement of quantum computing. QDataSet [Perrier et al., 2022] collects data from simulations of one- and two-qubit systems and targets training classical machine learning algorithms for quantum control, quantum tomography, and noise mitigation. LLM4QPE [Tang et al., 2023] is a large language model style paradigm for predicting quantum system properties with pre-training and fine-tuning workflows. While the paradigm is interesting, the empirical experiments are limited to two downstream tasks: quantum phase classification and correlation prediction. Fürrutter et al. [2024] studied the application of diffusion models [Sohl-Dickstein et al., 2015, Rombach et al., 2022] to quantum circuit synthesis [Saeedi and Markov, 2013, J. et al., 2022]. Although their methodology is appealing, scalability issues must be addressed to achieve practical and meaningful unitary compilation.

The aforementioned works represent meaningful explorations at the intersection of artificial intelligence and quantum computing. However, none of these datasets or models considers the task which

interests the quantum computing community (from the theoretical side) the most: quantum algorithm design. Our work aims to take the first step in bridging this gap. It is worth noting that several quantum algorithm benchmarks already exist, such as QASMBench [Li et al., 2023] and VeriQBench [Chen et al., 2022]. However, these benchmarks are designed to evaluate the performance of NISQ (Noisy Intermediate-Scale Quantum) [Preskill, 2018] machines, rather than for training and evaluating AI models. For instance, QASMBench includes a diverse variety of quantum circuits from different domains based on the OpenQASM assembly representation [Cross et al., 2022], covering quantum circuits with qubit sizes ranging from 2 to 127. However, each algorithm is represented by only 2-3 QASM files at most. While this is sufficient for benchmarking the fidelity of quantum hardware and the efficiency of QC compilers, it fails as a dataset for AI in that it does not capture the design patterns of each algorithm and ignores the construction of different oracles, which are crucial to quantum computing. Similar limitations apply to VeriQBench.

## 3 Preliminaries for Quantum Computing

In this section, we will introduce necessary backgrounds for quantum computing related to this paper. Additional preliminaries can also be found in Appendix B. A more detailed introduction to quantum computing can be found in the standard textbook by Nielsen and Chuang [2000].

**Quantum states.** In classical computing, the basic unit is a bit. In quantum computing, the basic unit is a *qubit*. Mathematically, $n$ ($n \in \mathbb{N}$) qubits forms an $N$-dimensional Hilbert space for $N = 2^n$. An $n$-qubit *quantum state* $|\phi\rangle$ can be written as

$$|\phi\rangle = \sum_{i=0}^{N-1} \alpha_i |i\rangle, \quad \text{where} \quad \sum_{i=0}^{N-1} |\alpha_i|^2 = 1. \tag{1}$$

Here $|\cdot\rangle$ represents a column vector, also known as a ket state. The tensor product of two quantum states $|\phi_1\rangle = \sum_{i=0}^{N-1} \alpha_i |i\rangle$ and $|\phi_2\rangle = \sum_{j=0}^{M-1} \beta_j |j\rangle$ with $M = 2^m$, $m \in \mathbb{N}$ is defined as

$$|\phi_1\rangle \otimes |\phi_2\rangle = \sum_{i=0}^{N-1} \sum_{j=0}^{M-1} \alpha_i \beta_j |i, j\rangle, \tag{2}$$

where $|i, j\rangle$ is an $(n + m)$-qubit state with first $n$ qubits being the state $|i\rangle$ and the last $m$ qubits being the state $|j\rangle$. When there is no ambiguity, $|\phi_1\rangle \otimes |\phi_2\rangle$ can be abbreviated as $|\phi_1\rangle|\phi_2\rangle$.

**Quantum oracles.** To study a Boolean function $f \colon \{0,1\}^n \to \{0,1\}^m$, we need to gain its access. Classically, a standard setting is to being able to *query* the function, in the sense that if we input an $x \in \{0,1\}^n$, we will get the output $f(x) \in \{0,1\}^m$. In quantum computing, the counterpart is a quantum query, which is instantiated by a *quantum oracle*. Specifically, the function $f$ is encoded as an oracle $U_f$ such that for any $x \in \{0,1\}^n$, $z \in \{0,1\}^m$,

$$U_f |x\rangle |z\rangle = |x\rangle |z \oplus f(x)\rangle, \tag{3}$$

where $\oplus$ is the plus modulo 2. Note that a quantum query to the oracle is stronger than a classical query in the sense that the quantum query can be applied to a state in *superposition*: For an input state $\sum_i c_i |x_i\rangle |z_i\rangle$ with $\sum_i |c_i|^2 = 1$, the output state is $\sum_i c_i |x_i\rangle |z_i \oplus f(x_i)\rangle$; measuring this state gives $x_i$ and $z_i \oplus f(x_i)$ with probability $|c_i|^2$. A classical query for $x$ can be regarded as the special setting with $c_1 = 1$, $x_1 = x$, $z_1 = 0^m$, and $c_i = 0$ for all other $i$.

**Quantum gates.** Similar to classical computing that can stem from logic synthesis with AND, OR, and NOT, quantum computing is also composed of basic quantum gates. For instance, the Hadamard $H$ is the matrix $\frac{1}{\sqrt{2}} \begin{bmatrix} 1 & 1 \\ 1 & -1 \end{bmatrix}$, satisfying $H|0\rangle = \frac{1}{\sqrt{2}}(|0\rangle + |1\rangle)$ and $H|1\rangle = \frac{1}{\sqrt{2}}(|0\rangle - |1\rangle)$. In general, an $n$-qubit quantum gate is a unitary matrix $\mathbb{C}^{2^n \times 2^n}$.

# 4 QCircuitNet Dataset

## 4.1 Task Suite

For the general purpose of quantum algorithm design, we consider two categories of tasks: oracle construction and algorithm design. These two tasks are crucial for devising and implementing a complete quantum algorithm, with oracle construction serving as the premise for algorithm design.

### 4.1.1 Task I: Oracle Construction

The construction of such an oracle $U_f$ using quantum gates is deeply rooted in the research topic of reversible quantum logic synthesis, which remains a challenge for complex Boolean functions. In this dataset, we mainly focus on the construction of textbook-level oracles: Bernstein-Vazirani Problem [Bernstein and Vazirani, 1993], Deutsch-Jozsa Problem [Deutsch and Jozsa, 1992], Simon's Problem [Simon, 1997], and Grover's algorithm for unstructured search [Grover, 1996] (including constructions of both the oracle and the diffusion operator). We also consider more advanced oracle construction tasks which we refer to as "Problem Encoding". For example, one can apply Grover's oracle to solving constraint problems such as SAT and triangle finding [Ambainis, 2004]. The intrinsic nature of formulating problem encoding tasks for LLMs slightly differs from quantum logic synthesis, and we refer the readers to Appendix B for more detailed discussion.

### 4.1.2 Task II: Quantum Algorithm Design

A general description of a quantum algorithm in natural language could be verbose and vague. Considering that quantum circuits stand at the core of designing and implementing a quantum algorithm, and that they resemble a special type of "language", we decide to use quantum circuits as the main medium for LLMs to generate for algorithm design. There are certain crucial points to consider when designing this framework to formulate the task precisely:

- From the perspective of quantum algorithm design, the oracle is usually provided as a blackbox gate since the goal of many algorithms is to determine the property of the function $f(x)$ encoded by the oracle $U_f$. If the model has access to the gate implementation of the oracle, it can directly deduce the property from the circuit, failing the purpose of designing a quantum algorithm to decode the information. However, for all experiment platforms, a quantum circuit needs to be explicitly constructed to compile and run successfully, which means the oracle should be provided with exact gate implementation. Most tutorials and benchmarks (especially those based on OpenQASM) simply merge the circuit implementation of the oracle and the algorithm as a whole for demonstration purposes. In our task of gearing LLMs for quantum algorithm design, how to separate the algorithm circuits from oracle implementation to avoid information leakage is a critical point to consider.

- A quantum algorithm constitutes not only the quantum circuit, but also the interpretation of execution (typically measurement) results of the quantum circuit. For example, in Simon's algorithm, the measurement results $y_i$ are not direct answer $s$ to the problem, but rather satisfies the property of $s \cdot y_i = 0$. Linear equations need to be solved to obtain the final answer. In this case, for a complete algorithm design, the model should also specify the way to process the execution results to derive the answer to the original problem.

- Quantum circuits for the same algorithm vary with different qubit number $n$. Although this is trivial for theoretical design, it needs to be considered when implementing concrete quantum circuits.

Beyond quantum algorithm design, we also consider quantum teleportation and quantum key distribution, since these protocols are widely used in quantum information. We cover their details in Appendix B.

## 4.2 Dataset Structure

The overall structure of QCircuitNet is illustrated as follows (more details are given in Appendix A):

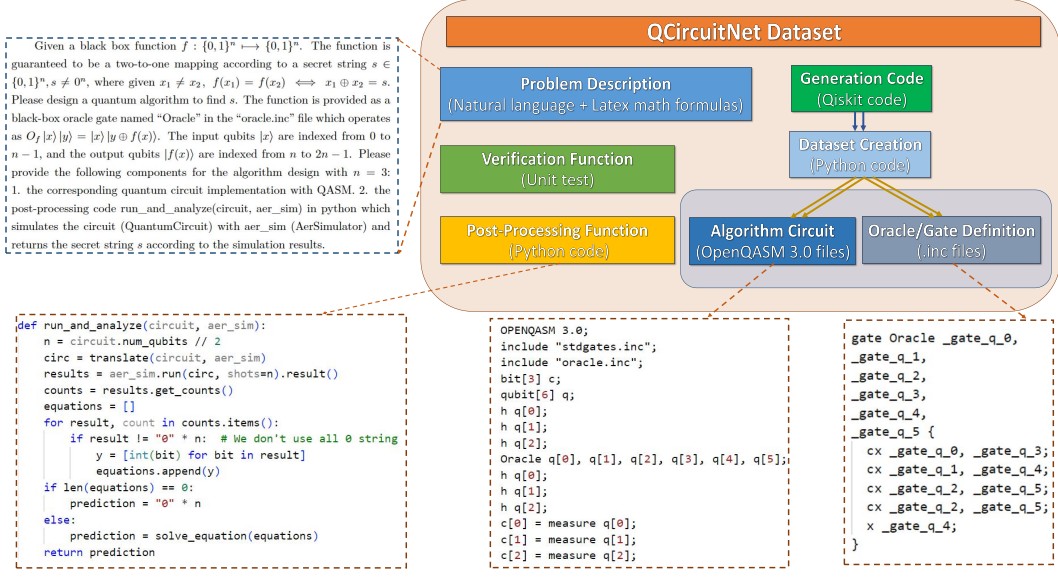

Figure 1: Structure of QCircuitNet. The components of QCircuitNet are presented in the frame on the top-right. As a showcase, this figure presents the components for Simon's problem [Simon, 1997], including its problem description in natural language, post-processing function in python code, circuit in a .qasm file, and oracle definition in a .inc file.

**Design Principles.** As discussed in Section 4.1, a critical consideration in formulating the framework is the dilemma between providing the oracle as a black box for quantum algorithm design and the need for its explicit construction to execute the circuit and interpret the results, making the algorithm design complete. Additionally, model training and reference present challenges, particularly for LLMs in generating complex and precise composite gates and evaluating the results efficiently. To address these obstacles, we highlight the following construction principles, which are specially designed to adapt to these two tasks:

• For algorithm design tasks, as discussed in Section 4.1.2, we provide the oracle as a black-box gate named "Oracle" with the explicit definition in a separate "oracle.inc" library, which is supported by the OpenQASM 3.0 grammar. In this way, we make sure that the model can use the oracle without accessing its underlying function, which solves the problem of isolating oracle definition from the algorithm circuit.

• For oracle construction tasks, we ask the model to directly output the quantum circuit in QASM format. For algorithm design task, we require both a quantum circuit and a post-processing function to derive the final answer from circuit execution results. Moreover, we ask the model to explicitly set the shots needed to run the circuit itself in order to characterize the query complexity, which is critical in the theoretical analysis of algorithms.

• For available quantum gates, we provide the definition of some important composite gates not included in the standard QASM gate library in a "customgates.inc". Hierarchical definition for multi-controlled X gate contains 45060 lines for qubit number $n = 14$ in OpenQASM format, which is impossible for AI models to accurately generate at the time. Providing these as a .inc file guarantees the correctness of OpenQASM's grammar while avoiding the generation of complicated gates, which is a distraction from the original design task.

• To verify models' output automatically without human evaluation, we compose verification functions to validate the syntax of QASM / Qiskit and the functionality of the implemented circuits / codes. Since comprehensive Logic Equivalence Checking (LEC) might be inefficient for the throughput of LLM inference, we perform the verification by directly checking the correctness of output with extensive test cases.

Based on theses principles, we proposed the framework of QCircuitNet. Below is a more detailed explanation for the 7 components of the dataset:

1. **Problem Description:** carefully hand-crafted prompts stating the oracle to be constructed or the target problem to be solved in natural language and latex math formulas. If the problem involves the usage of a quantum oracle or composite gates beyond the standard gate library, the interfaces of the oracle / gate will also be included (input qubits, output qubits, function mechanism).

2. **Generation Code:** one general Qiskit [Javadi-Abhari et al., 2024] code to create quantum circuits for oracles or algorithms of different settings, such as distinct secret strings or various qubit numbers. We choose Qiskit as the main experiment platform because it is a general quantum programming software widely used for the complete workflow from creating quantum circuits to transpiling, simulation, and execution on real hardware.

3. **Algorithm Circuit:** a .qasm file storing the quantum circuit for each specific setting. We choose OpenQASM 3.0 [Cross et al., 2022] as the format to store the quantum circuits, because Qiskit, as a python library, can only create quantum circuits at runtime instead of explicitly saving the circuits at gate level.[1]

4. **Post-Processing Function:** this is for Algorithm Design task only, see Section 4.1.2. The function takes a complete quantum circuit as input, uses the Qiskit AerSimulator to execute the circuit, and returns the final answer to the original problem according to the simulation results. For state preparation problems such as creating a GHZ state of $n$ qubits, this function returns the qubit indices of the generated state.

5. **Oracle / Gate Definition:** a .inc file to provide definitions of composite gates or oracles. For oracle construction tasks, this only includes the definition of composite gates required to build the oracle. For algorithm design tasks, we also provide the gate definition of the oracle in this file, which successfully delivers the oracle in a black-box way.

6. **Verification Function:** a function to evaluate whether the implemented oracle / algorithm successfully achieves the desired purpose with grammar validation and test cases verification. The function returns -1 if there exist grammar errors, and returns a score between $[0, 1]$ indicating the success rate on test cases.[2]

7. **Dataset Creation Script:** the script to create the dataset from scratch in the format suitable for fine-tuning / evaluating LLMs. It contains the following functions: 1. generate primitive QASM circuits. 2. extract gate definitions and add include instructions to create algorithm circuit, the direct output of model. 3. validate and verify the correctness of the data points in the dataset. 4. concatenate algorithm circuit with problem description as a json file for the benchmark pipeline.

This structure of QCircuitNet provides a general framework to formulate quantum algorithm design for large language models, with an easy extension to more advanced quantum algorithms.

# 5 Experiments

## 5.1 Methodology for Benchmarking

We benchmark the quantum algorithm design capabilities of leading closed-source and open-source large language models using QCircuitNet. The workflow of our benchmark is illustrated in Figure 2. The total computation cost is approximately equivalent to two days on an A100 GPU.

---

[1]Although currently the Qiskit APIs for importing and dumping OpenQASM 3.0 files are still in experimental stage, we choose to adopt version 3.0 over 2.0 in that it supports parameterized circuits, which allows for extending the framework to variational quantum algorithms [Cerezo et al., 2021] by saving parameterized varational ansatzes.

[2]The verification function explicitly integrates the oracle / gate definition library with output algorithm circuit since Qiskit importer for OpenQASM 3.0 does not support non-standard gate libraries currently.

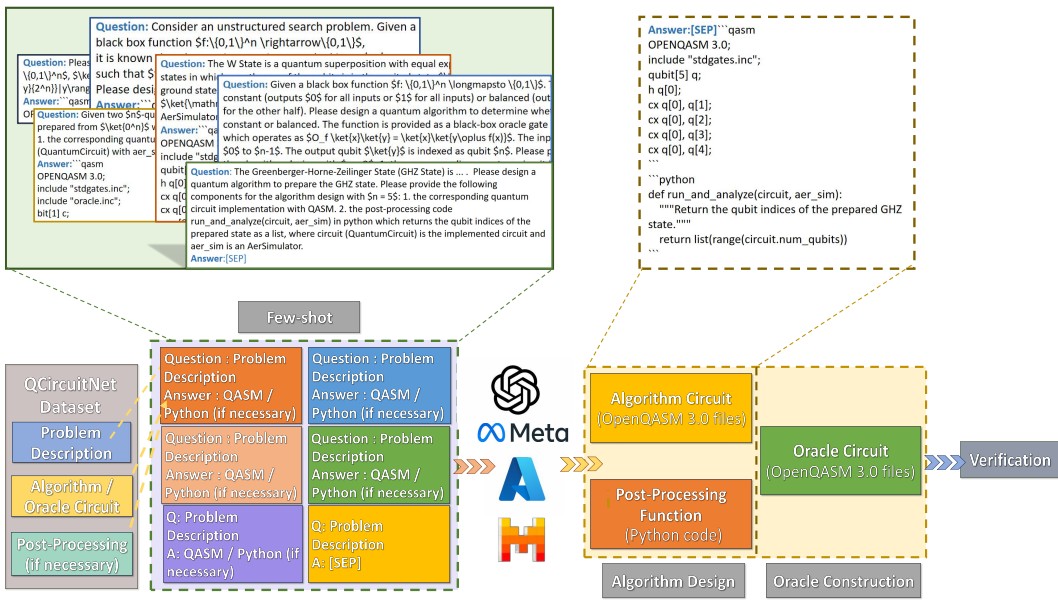

Figure 2: Flowchart of benchmarking QCircuitNet.

**Models.** Recently, the GPT series models have become the benchmark for generative models due to their exceptional performance. Specifically, we include two models from OpenAI, GPT-3.5-turbo [Brown et al., 2020] and GPT-4 [OpenAI et al., 2024], in our benchmark. Additionally, the LLAMA series models [Touvron et al., 2023a,b] are widely recognized as leading open-source models, and we have selected LLAMA-3-8B for our study. For a comprehensive evaluation, we also benchmark Phi-3-medium-128k [Abdin et al., 2024] and Mistral-7B-v0.3 [Jiang et al., 2023].

**Prompts.** We employ a few-shot learning framework, a prompting technique that has shown considerable success in generative AI [Xie et al., 2021]. In this approach, we utilize either 1 or 5 examples, followed by a problem description. To ensure we do not train and test on the same quantum algorithm, we implement k-fold validation. This method involves using one problem as the test set while the remaining problems serve as the training set, rotating through each problem one at a time.

**Evaluation Metrics.** We use three evaluation metrics:

1. BLEU Score: this metric measures how closely the generated code matches the reference code, with a higher BLEU score indicating greater similarity.

2. Byte Perplexity: this metric evaluates the model's ability to predict the next byte in a sequence. Lower byte perplexity indicates better performance by reflecting the model's predictive accuracy.

3. Verification function: this function checks the syntax validation and the result correctness of the code produced by the language model, and returns a score depending on the performance. See Section 4.2 for more detailed discussion.

### 5.2 Results

The results for BLEU and verification function score are shown in Figure 3, Table 1, and Table 2. We include the results of Byte Perplexity and more experiments in Appendix C.

As illustrated in the table, verification scores for the output of the model reveal that almost none can produce a correct algorithm, because a single mistake could make the whole algorithm fail. However, we can still partially assess the models' ability to solve quantum problems by measuring the BLEU

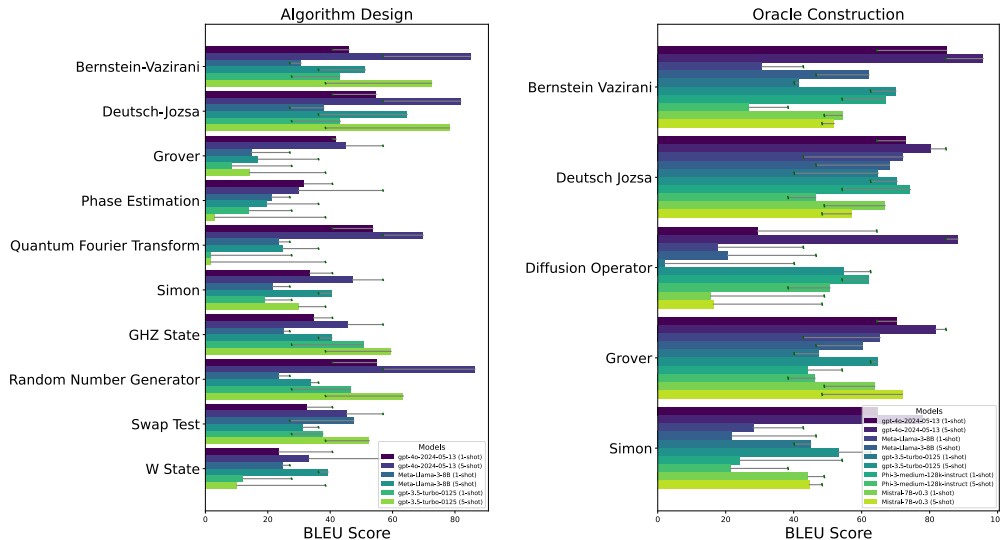

Figure 3: Benchmarking algorithm design and oracle construction in BLEU scores.

score. The figure indicates that GPT-4o significantly outperforms all other models. Additionally, nearly all models demonstrate the ability to learn quantum knowledge from context, as the five-shot prompt performs much better than the one-shot alternative.

The figure also reveals the different difficulty levels for each algorithm. For simple quantum algorithms such as the Bernstein-Vazirani algorithm where directly applying more H gates to the qubits solves the problem, language models tend to perform well. However, for complicated algorithms such as the W state where the parameters vary with qubit number, the models tend to perform poorly.

Table 1: Benchmarking algorithm design in verification function scores.

| Model | Shot | Bernstein-Vazirani | Deutsch-Jozsa | Grover | Phase Estimation | Quantum Fourier Transform | Simon | GHZ State | Random Number Generator | Swap Test | W State |
|---|---|---|---|---|---|---|---|---|---|---|---|
| gpt-4o-2024-05-13 | 1 | -1 | -1 | -1 | -1 | -1 | -1 | -0.846153846 | -1 | -1 | -1 |
| gpt-4o-2024-05-13 | 5 | -1 | -1 | -1 | -1 | -1 | -1 | -0.153846154 | 0.405072709 | -1 | -0.846153846 |
| Meta-Llama-3-8B | 1 | -1 | -1 | -1 | -1 | -1 | -1 | -0.769230769 | -0.928534157 | -1 | -0.461538462 |
| Meta-Llama-3-8B | 5 | -1 | -1 | -1 | -1 | -1 | -1 | -0.384615385 | -0.730665436 | -1 | -0.153846154 |
| gpt-3.5-turbo-0125 | 1 | -1 | -1 | -1 | -1 | -1 | -1 | -0.846153846 | -1 | -1 | -1 |
| gpt-3.5-turbo-0125 | 5 | -1 | -1 | -1 | -1 | -1 | -1 | -0.076923077 | -0.490434406 | -1 | -0.846153846 |

Table 2: Benchmarking oracle construction in verification function scores.

| Model | Shot | Bernstein-Vazirani | Deutsch-Jozsa | Diffusion-Operator | Grover | Simon |
|---|---|---|---|---|---|---|
| gpt-4o-2024-05-13 | 1 | 0.15 | 0.22 | -0.923076923 | -0.977011494 | -0.260869565 |
| gpt-4o-2024-05-13 | 5 | 0.15 | 0.43 | -0.230769231 | -0.931034483 | -0.043478261 |
| Meta-Llama-3-8B | 1 | -0.64 | -0.49 | -0.615384615 | -1 | -0.456521739 |
| Meta-Llama-3-8B | 5 | -0.06 | 0.21 | -0.615384615 | -1 | -0.423913043 |
| gpt-3.5-turbo-0125 | 1 | -0.4 | -0.01 | -0.846153846 | -0.977011494 | -0.423913043 |
| gpt-3.5-turbo-0125 | 5 | -0.07 | 0.06 | -0.307692308 | -0.896551724 | -0.108695652 |
| Phi-3-medium-128k-instruct | 1 | -0.5 | -0.52 | -0.846153846 | -1 | -0.673913043 |
| Phi-3-medium-128k-instruct | 5 | -0.6 | -0.22 | -1 | -1 | -0.760869565 |
| Mistral-7B-v0.3 | 1 | -0.35 | -0.47 | -1 | -1 | -0.369565217 |
| Mistral-7B-v0.3 | 5 | -0.11 | -0.02 | -1 | -1 | -0.217391304 |

## 5.3 Observations and Analysis

**The Challenge of LLM for Quantum Algorithm Design.** As shown by the experiment results, the integration of LLMs into quantum algorithm design presents several challenges:

1. Lack of data: Unlike classical computing and code generation, where vast datasets and extensive examples exist, the field of quantum computing is still nascent, with limited accessible data. This scarcity hampers the ability of LLMs to learn and generalize effectively.

2. Distinct nature of each algorithm: Quantum algorithms can be seen as unitary maps but in exponential size linear spaces. This distinct nature makes it intractable for LLMs to generalize knowledge from one algorithm to another, posing challenges to transfer learning.

3. Reasoning of underlying mechanism: Quantum algorithms involve deep comprehension of unitary transformations and the evolution of quantum states. Such reasoning goes beyond simple pattern recognition and is difficult for LLMs to grasp and apply accurately.

4. Quantum programming language syntax: The syntax of quantum programming languages, such as Qiskit and OpenQASM, introduces an additional layer of complexity. As shown by the verification scores, the models can barely output circuit / codes with correct syntax, demonstrating that this is a non-trivial task, which challenges the current capabilities of LLMs.

**Usage of QCircuitNet Dataset.**   Our dataset helps provide guidance to address these challenges:

1. Formulate the task: We propose framing algorithm design tasks in circuit or code form rather than natural language descriptions, which can be vague, or mathematical formulas, which are difficult to verify. This provides a concrete framework for LLMs to operate within.

2. Clarify descriptions with concrete examples: The dataset includes detailed descriptions of representative problems in universal quantum algorithms, accompanied by concrete cases, which helps bridge the gap between abstract algorithms and practical implementations.

3. Benchmark for fair evaluation: To improve the capability of LLMs in quantum algorithm design, we need a fair and robust evaluation method first. Our dataset includes metrics and benchmarks for such purpose, providing a foundation for developing and testing novel improvement methods.

**Implications for AI Learning.**   We observe a performance separation between writing general qiskit codes and explicit gate-level circuits in QASM. Since Qiskit provides detailed tutorial with general codes for several algorithms, this may imply a *data contamination* phenomenon where LLMs rely on memorization and retrieval rather than genuine algorithm design. Similarly, current benchmarks for AI code generation and syntax learning may also suffer from this unseen bias. Our dataset, based on QASM files created from scratch, may help circumvent this issue and serve as a stable and fair evaluation method for benchmarking AI syntax learning.

# 6   Conclusions and Future Work

In this paper, we propose QCircuitNet, the first comprehensive, structured universal quantum algorithm dataset and quantum circuit generation benchmark for AI models. It contains automatic validation and verification functions, allowing for scalable and efficient evaluation methodologies. Benchmarking of QCircuitNet on up-to-date LLMs are systematically conducted.

Our work leaves several open questions for future investigation:

• QCircuitNet is a benchmarking dataset for LLMs. It is of general interest to extend benchmarking to training, which will help LLMs better maneuver quantum algorithm design. This may need implementations of more advanced algorithms to make it a more meaningful training dataset.

• Since quantum algorithms have fundamental difference from classical algorithms, novel fine-tuning methods to attempt quantum algorithm design and quantum circuit implementation, or even development of new quantum algorithms by LLMs are solicited.

• Currently, variational quantum algorithms [Cerezo et al., 2021] can already be implemented on near-term NISQ machines [Preskill, 2018]. It would be also of general interest to extend QCircuitNet to contain the design and implementation of variational quantum algorithms.

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
