| gpt-4o-2024-05-13 | 5 | -0.1100 | 0.0800 | -0.3077 | -0.9540 | -0.0870 |
| Meta-Llama-3-8B | 1 | -0.7300 | -0.5000 | -0.3846 | -1.0000 | -0.6848 |
| Meta-Llama-3-8B | 5 | -0.0500 | 0.1700 | -0.8462 | -1.0000 | -0.6413 |
| gpt-3.5-turbo-0125 | 1 | -0.3500 | -0.0400 | -0.8462 | -1.0000 | -0.3696 |
| gpt-3.5-turbo-0125 | 5 | -0.1100 | 0.0200 | -0.3077 | -0.9770 | -0.1087 |
| Phi-3-medium-128k-instruct | 1 | -0.6800 | -0.6100 | -0.9231 | -1.0000 | -0.7500 |
| Phi-3-medium-128k-instruct | 5 | -0.5400 | -0.4300 | -1.0000 | -1.0000 | -0.8370 |
| Mistral-7B-v0.3 | 1 | -0.4000 | -0.4300 | -0.9231 | -0.9540 | -0.6087 |
| Mistral-7B-v0.3 | 5 | -0.3700 | -0.1300 | -1.0000 | -0.9195 | -0.2391 |

## 5.3 Observations and Analysis

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

## A Details of QCircuitNet

The QCircuitNet Dataset, along with its Croissant metadata, is available on Anonymous GitHub at the following link: https://anonymous.4open.science/r/QCircuitNet-DE28/

QCircuitNet has the following directory structure:

```
QCircuitNet
├── Oracle Construction ..................... All data for the oracle construction task
│   ├── Quantum Logic Synthesis ........ Textbook-level oracles used in the experiments
│   ├── Problem Encoding .............. Advanced oracles encoding application scenarios
├── Algorithm Design ................... All data for the quantum algorithm design task
    ├── Quantum Computing ....... Textbook-level universal quantum computing algorithms
    ├── Quantum Information ..... Textbook-level quantum information tasks and protocols
```

In each subdirectory, there is a folder for each specific algorithm. For instance, the folder structure for Simon's algorithm is as follows:

```
Algorithm Design
├── Quantum Computing
    ├── simon ....................................... All data for the Simon's Problem
        ├── simon-dataset.py ................................. Dataset creation script
        ├── simon-generation.py ............................. Qiskit generation code
        ├── simon-post-processing.py ...................... Post-processing function
        ├── simon-utils.py .......................... Utility functions for verification
        ├── simon-verification.py ............................. Verification function
        ├── simon-description.txt ............................. Problem description
        ├── simon-verification.txt ............... Verification results of the data points
        ├── full circuit ............................... Raw data of quantum circuits
        │   ├── simon-n2
        │   │   ├── simon-n2-s11-k11.qasm .............. Full circuit for a concrete setting
        │   ├── simon-n3
        │   │   ├── simon-n3-s011-k001.qasm
        │   │   ├── simon-n3-s011-k101.qasm
        │   │   ├── simon-n3-s100-k001.qasm
        │   │   ├── simon-n3-s100-k101.qasm
        │   ├── ...
        ├── test oracle .................................. Extracted oracle definitions
        │   ├── n2
        │   │   ├── trial1
        │   │   │   ├── oracle.inc .......................... Oracle definition as a .inc file
        │   │   │   ├── oracle-info.txt ........... Oracle information (such as key strings)
        │   ├── n3
        │   │   ├── trial1
        │   │   │   ├── oracle.inc
        │   │   │   ├── oracle-info.txt
        │   │   ├── trial2
        │   │   │   ├── oracle.inc
        │   │   │   ├── oracle-info.txt
        │   │   ├── trial3
        │   │   │   ├── oracle.inc
        │   │   │   ├── oracle-info.txt
        │   │   ├── trial4
        │   │   │   ├── oracle.inc
        │   │   │   ├── oracle-info.txt
        │   ├── ...
        ├── simon-n2.qasm ......................... Algorithm circuit for model output
        ├── simon-n3.qasm
        ├── simon-n4.qasm
        ├── simon-n5.qasm
        ├── ...
```

538   We expect to extend QCircuitNet under this general structure.

## A.1   Format

540   In this subsection, we provide concrete examples to illustrate the different components of QCircuitNet.
541   We use the case of Simon's Problem throughout the demonstration to achieve better consistency. For
542   further details, please check the code repository.

543   1. **Problem Description:** this is the carefully hand-crafted description of the task in natural language
544      and latex math formulas. The description is provided as one template for each algorithm, and the
545      concrete settings (such as the qubit number) are replaced when creating the data points in json.
546      The file is named as "{algorithm_name}_description.txt".

---

**Problem Description Template for Simon's Problem**

Given a black box function $f : \{0,1\}^n \longmapsto \{0,1\}^n$. The function is guaranteed to be a two-to-one mapping according to a secret string $s \in \{0,1\}^n, s \neq 0^n$, where given $x_1 \neq x_2, f(x_1) = f(x_2) \iff x_1 \oplus x_2 = s$. Please design a quantum algorithm to find $s$. The function is provided as a black-box oracle gate named "Oracle" in the "oracle.inc" file which operates as $O_f |x\rangle |y\rangle = |x\rangle |y \oplus f(x)\rangle$. The input qubits $|x\rangle$ are indexed from 0 to $n-1$, and the output qubits $|f(x)\rangle$ are indexed from $n$ to $2n-1$. Please provide the following components for the algorithm design with $n =$ {qubit number}: 1. the corresponding quantum circuit implementation with {QASM / Qiskit}. 2. the post-processing code run_and_analyze(circuit, aer_sim) in python which simulates the circuit (QuantumCircuit) with aer_sim (AerSimulator) and returns the secret string $s$ according to the simulation results.

---

547

548   2. **Generation Code:** one general Qiskit code to create quantum circuits of different settings. Note
549      that the oracle for the problem is provided as a black-box gate "oracle" here. This code is used to
550      generate the raw data, but can also be used as a testing benchmark for writing Qiskit codes. The
551      file is named as "{algorithm_name}_generation.py".

```python
from qiskit import QuantumCircuit

def simon_algorithm(n, oracle):
    """Generates a Simon algorithm circuit.

    Parameters:
    - n (int): number of qubits
    - s (str): the secret string of length n

    Returns:
    - QuantumCircuit: the Simon algorithm circuit
    """
    # Create a quantum circuit on 2n qubits
    simon_circuit = QuantumCircuit(2 * n, n)

    # Initialize the first register to the |+> state
    simon_circuit.h(range(n))

    # Append the Simon's oracle
    simon_circuit.append(oracle, range(2 * n))

    # Apply a H-gate to the first register
    simon_circuit.h(range(n))

    # Measure the first register
    simon_circuit.measure(range(n), range(n))

    return simon_circuit
```

Listing 1: Qiskit generation code for Simon's algorithm.

3. **Algorithm Circuit:** the OpenQASM 3.0 format file storing the quantum circuit in gate level for each specific setting. Note that the explicit construction of "Oracle" is provided separately in "oracle.inc" file, which guarantees the usage of oracle in a black-box way. This filed is named as "{algorithm_name}_n{qubit_number}.qasm".

```
OPENQASM 3.0;
include "stdgates.inc";
include "oracle.inc";
bit[3] c;
qubit[6] q;
h q[0];
h q[1];
h q[2];
Oracle q[0], q[1], q[2], q[3], q[4], q[5];
h q[0];
h q[1];
h q[2];
c[0] = measure q[0];
c[1] = measure q[1];
c[2] = measure q[2];
```

Listing 2: OpenQASM 3.0 Code for Simon's algorithm with $n = 3$.

4. **Post-Processing Function:** this function simulates the quantum circuit and derives the final answer to the problem. The file is named as "{algorithm_name}_post_processing.py".

```python
from sympy import Matrix
import numpy as np
from qiskit import transpile

def mod2(x):
    return x.as_numer_denom()[0] % 2

def solve_equation(string_list):
    """
    A^T | I
    after the row echelon reduction, we can get the basis of the
        ↪ nullspace of A in I
    since we just need the string in binary form, so we can just
        ↪ use the basis
    if row == n-1 --> only one
    if row < n-1 --> get the first one (maybe correct or wrong)
    """
    M = Matrix(string_list).T

    # Augmented   : M | I
    M_I = Matrix(np.hstack([M, np.eye(M.shape[0], dtype=int)]))

    # RREF row echelon form , indices of the pivot columns
    # If x % 2 = 0, it will not be chosen as pivot (modulo 2)
    M_I_rref = M_I.rref(iszerofunc=lambda x: x % 2 == 0)

    # Modulo 2
    M_I_final = M_I_rref[0].applyfunc(mod2)

    # Non-Trivial solution
    if all(value == 0 for value in M_I_final[-1, : M.shape[1]]):
        result_s = "".join(str(c) for c in M_I_final[-1, M.shape[1]
            ↪   :])
```

```
      # Trivial solution
      else:
          result_s = "0" * M.shape[0]

      return result_s

def run_and_analyze(circuit, aer_sim):
    n = circuit.num_qubits // 2
    circ = transpile(circuit, aer_sim)
    results = aer_sim.run(circ, shots=n).result()
    counts = results.get_counts()
    equations = []
    for result, count in counts.items():
        if result != "0" * n:  # We don't use all 0 string
            y = [int(bit) for bit in result]
            equations.append(y)
    if len(equations) == 0:
        prediction = "0" * n
    else:
        prediction = solve_equation(equations)
    return prediction
```

Listing 3: Post-processing code for Simon's algorithm.

5. **Oracle / Gate Definition:** this .inc file provides the definitions of composite gates or oracles. The file is named "customgates.inc" for oracle construction tasks and "oracle.inc" for algorithm design tasks.

```
gate Oracle _gate_q_0, _gate_q_1, _gate_q_2, _gate_q_3, _gate_q_4,
    ↪  _gate_q_5 {
  cx _gate_q_0, _gate_q_3;
  cx _gate_q_1, _gate_q_4;
  cx _gate_q_2, _gate_q_5;
  cx _gate_q_2, _gate_q_5;
  x _gate_q_3;
}
```

Listing 4: One test case oracle for Simon's algorithm with $n = 3$.

For algorithm design tasks, this .inc file is accompanied with an "oracle_info.txt" file to describe the encoded information of the oracle. This helps the verification function to check the correctness of the derived answer by the model. The above test case is equipped with the following information text:

> oracle_info.txt for Simon's Problem with qubit number 3 and test case 2.
>
> Secret string: 100
> Key string: 001

6. **Verification Function:** the function to evaluate the output with grammar validation and test cases verification. The file is named as "{algorithm_name}_verification.py".

```
from simon_utils import *

def check_model(qasm_string, code_string, n):
    """Check the Simon model."""
    # Verify the syntax of the QASM code with the first test case
    ↪  oracle
    t = 1
    with open(f"test_oracle/n{n}/trial{t}/oracle.inc", "r") as file
        ↪  :
```

```python
            oracle_def = file.read()
        full_qasm = plug_in_oracle(qasm_string, oracle_def)
        circuit = verify_qasm_syntax(full_qasm)
        if circuit is None:
            return -1
        try:
            exec(code_string, globals())
            aer_sim = AerSimulator()
            total_success = 0
            total_fail = 0
            t_range = min(10, 4 ** (n - 2))
            shots = 10
            for t in range(1, 1 + t_range):
                print(f"    Running Test Case {t}")
                with open(f"test_oracle/n{n}/trial{t}/oracle.inc", "r")
                    ↪    as file:
                    oracle_def = file.read()
                full_qasm = plug_in_oracle(qasm_string, oracle_def)
                circuit = loads(full_qasm)
                with open(f"test_oracle/n{n}/trial{t}/oracle_info.txt",
                    ↪    "r") as file:
                    content = file.read()
                match = re.search(r"Secret string: ([01]+)", content)
                if match:
                    secret_string = match.group(1)
                else:
                    raise ValueError("Secret string not found in the
                        ↪    file.")

                cnt_success = 0
                cnt_fail = 0
                for shot in range(shots):
                    prediction = run_and_analyze(circuit.copy(),
                        ↪    aer_sim)
                    if not isinstance(prediction, str):
                        raise TypeError("Predicted secret string should
                            ↪    be a string.")
                    if prediction == secret_string:
                        cnt_success += 1
                    else:
                        cnt_fail += 1
                print(f"        Success: {cnt_success}/{shots}, Fail: {
                    ↪    cnt_fail}/{shots}")
                total_success += cnt_success
                total_fail += cnt_fail
            print(f"Total Success: {total_success}; Total Fail: {
                ↪    total_fail}")
            return total_success / (total_fail + total_success)

        except Exception as e:
            print(f"Error: {e}")
            return -1
```

Listing 5: Verification function for Simon's algorithm.

This verification function is accompanied with an "{algorithm_name}_utils.py" file to provide necessary utility functions.

```python
from qiskit.qasm3 import loads
from qiskit_aer import AerSimulator
import re

def print_and_save(message, text):
    print(message)
```

```
        text.append(message)

def plug_in_oracle(qasm_code, oracle_def):
    """Plug-in the oracle definition into the QASM code."""
    oracle_pos = qasm_code.find('include "oracle.inc";')
    if oracle_pos == -1:
        raise ValueError("Oracle include statement not found in the
        ↪   file")
    full_qasm = (
        qasm_code[:oracle_pos]
        + oracle_def
        + qasm_code[oracle_pos + len('include "oracle.inc";') :]
    )
    return full_qasm

def verify_qasm_syntax(output):
    """Verify the syntax of the output and return the corresponding
    ↪   QuantumCircuit (if it is valid)."""
    assert isinstance(output, str)
    try:
        # Parse the OpenQASM 3.0 code
        circuit = loads(output)
        print(
            "    The OpenQASM 3.0 code is valid and has been
            ↪ successfully loaded as a QuantumCircuit."
        )
        return circuit
    except Exception as e:
        print(f"    Error: The OpenQASM 3.0 code is not valid.
        ↪ Details: {e}")
        return None
```

Listing 6: Utility functions for verification of Simon's algorithm.

7. **Dataset Creation Script:** this script involves all the code necessary to create the data points from scratch. The file is named as "{algorithm_name}_dataset.py". The main function looks like this:

```
def main():
    parser = argparse.ArgumentParser()
    parser.add_argument(
        "-f",
        "--func",
        choices=["qasm", "json", "gate", "check"],
        help="The function to call: generate qasm circuit, json
            ↪ dataset or extract gate definition.",
    )
    args = parser.parse_args()
    if args.func == "qasm":
        generate_circuit_qasm()
    elif args.func == "json":
        generate_dataset_json()
    elif args.func == "gate":
        extract_gate_definition()
    elif args.func == "check":
        check_dataset()
```

Listing 7: Main function of the dataset script for Simon's algorithm.

Here the "generate_circuit_qasm()" function generates the raw data of quantum circuits in Open-QASM 3.0 format where the algorithm circuit and the oracle definition are blended, then "extract_gate_definition()" function extracts the definition of oracles and formulates the algorithm circuits into the format suitable for model output. The "check_dataset()" function is used to check

the correctness of the created data points and "generate_dataset_json()" function to combine the data into json format for easy integration with the benchmarking pipeline.

## A.2  Discussion of more Tasks

**Problem Encoding.**    In Section 4.1.1, we mentioned another category of oracle construction tasks referred to as "Problem Encoding", which involves applying quantum algorithms, such as Grover's algorithm, to solve practical problems such as SAT and triangle finding. The crux of this process is encoding the problem constraints into Grover's oracle, thereby making this a type of oracle construction task. Unlike quantum logic synthesis, which encodes an explicit function $f(x)$ as a unitary operator $U_f$, this task involves converting the constraints of a particular problem into the required oracle form. We provide implementations of several concrete problems in this directory as demonstrations and will include more applications in future work.

**Quantum Information Protocols.**    In the "Quantum Information" section of the "Algorithm Design" task, we have also implemented three important quantum information protocols: Quantum Teleportation, Superdense Coding, and Quantum Key Distribution (BB84). A brief introduction to these protocols can be found in Appendix B. We did not include the experiments for these protocols as they involve communication between two parties, which is challenging to characterize with a single OpenQASM 3.0 file. We recommend revising the post-processing function as a general classical function to schedule the communication and processing between different parties specifically for these protocols. The fundamental quantum circuits and processing codes are provided in the repository.

## A.3  Datasheet

Here we present a datasheet for the documentation of QCircuitNet.

**Motivation**

- *For what purpose was the dataset created?* It was created as a benchmark for the capability of designing and implementing quantum algorithms for LLMs.

- *Who created the dataset (e.g., which team, research group) and on behalf of which entity (e.g., company, institution, organization)?* The authors of this paper.

- *Who funded the creation of the dataset?* We will reveal the funding resources in the Acknowledgement section of the final version.

**Composition**

- *What do the instances that comprise the dataset represent (e.g., documents, photos, people, countries)?* The dataset comprises problem description, generation code, algorithm circuit, post-processing function, oracle / gate definition, verification function, and dataset creation script for various quantum algorithms.

- *How many instances are there in total (of each type, if appropriate)?* The dataset has 5 algorithms for oracle construction task and 10 algorithms for algorithm design task used for experiments. There are 3 quantum information protocols and additional problem encoding tasks not included for experiments.

- *Does the dataset contain all possible instances or is it a sample (not necessarily random) of instances from a larger set?* The dataset contains instances with restricted qubit numbers due to the current scale of real quantum hardware.

- *What data does each instance consist of?* Qiskit codes, OpenQASM 3.0 codes, python scripts, and necessary text information.

- *Are relationships between individual instances made explicit?* Yes, the way to create different instances are clearly described in Appendix A.1.

- *Are there recommended data splits?* Yes, we recommend splitting the data according to different algorithms in algorithm design task.

- *Are there any errors, sources of noise, or redundancies in the dataset?* There might be some small issues due to the dumping process of Qiskit and programming mistakes (if any).

- *Is the dataset self-contained, or does it link to or otherwise rely on external resources (e.g., websites, tweets, other datasets)?* The dataset is self-contained.

- *Does the dataset contain data that might be considered confidential (e.g., data that is protected by legal privilege or by doctor-patient confidentiality, data that includes the content of individuals' non-public communications)?* No.

- *Does the dataset contain data that, if viewed directly, might be offensive, insulting, threatening, or might otherwise cause anxiety?* No.

**Collection Process**

- *How was the data associated with each instance acquired?* The data is created by first composing Qiskit codes for each algorithm and then converting to OpenQASM 3.0 files using qiskit.qasm3.dump function, with additional processing procedure.

- *What mechanisms or procedures were used to collect the data (e.g., hardware apparatuses or sensors, manual human curation, software programs, software APIs)?* Manual human programming and Qiskit APIs.

- *Who was involved in the data collection process (e.g., students, crowd workers, contractors), and how were they compensated (e.g., how much were crowd workers paid)?* Nobody other than the authors of the paper.

- *Over what timeframe was the data collected?* The submitted version of the dataset was created in May and June 2024.

**Uses**

- *Has the dataset been used for any tasks already?* It has been used in this paper to benchmark LLM's ability for quantum algorithm design.

- *Is there a repository that links to any or all papers or systems that use the dataset?* The only paper which uses the dataset for now is this paper.

**Distribution**

- *Will the dataset be distributed to third parties outside of the entity (e.g., company, institution, organization) on behalf of which the dataset was created?* Yes, the dataset will be made publicly available on the Internet after the review process.

- *How will the dataset be distributed (e.g., tarball on website, API, GitHub)?* It will be distributed on the GitHub platform.

- *Will the dataset be distributed under a copyright or other intellectual property (IP) license, and/or under applicable terms of use (ToU)?* The dataset is distributed under CC BY 4.0.

- *Have any third parties imposed IP-based or other restrictions on the data associated with the instances?* No.

- *Do any export controls or other regulatory restrictions apply to the dataset or to individual instances?* No.

**Maintenance**

- *Who will be supporting/hosting/maintaining the dataset?* The authors of this paper.

- *How can the owner/curator/manager of the dataset be contacted (e.g., email address)?* The email for contact will be provided after the review process.

- *Is there an erratum?* Not at this time.

- *Will the dataset be updated (e.g., to correct labeling errors, add new instances, delete instances)?* Yes, it will be continually updated.

- *If others want to extend/augment/build on/contribute to the dataset, is there a mechanism for them to do so?* Yes, they can do so with the GitHub platform.

### A.4 Copyright and Licensing Terms

This work is distributed under a CC BY 4.0 license. The implementation of the code references open-source projects such as Qiskit, QuantumKatas, Cirq, and NWQBench. We bear responsibility in case of violation of rights.

## B Additional Preliminaries for Quantum Computing and Quantum Information

**Quantum circuit diagram.** A quantum algorithm is composed of a series of quantum gates. By default, a quantum algorithm starts from the all-0 state $|0^n\rangle$. A quantum algorithm can be illustrated by its quantum gate diagram, drawn from left to right. The initial all-0 state is placed at the left side of the diagram. After that, whenever we apply a quantum gate, it is placed on the corresponding qubits, from left to right. At the end of the quantum gates, we need to measure and read the outputs, and these measurements are placed at the right side of the diagram. See Figure 4 for the quantum gate diagram of Simon's algorithm [Simon, 1997].

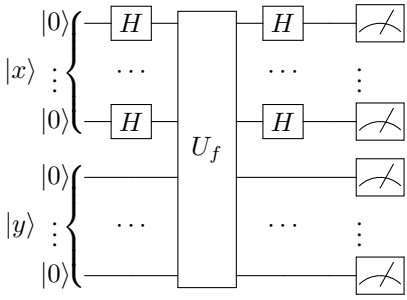

Figure 4: Quantum gate diagram of Simon's algorithm.

**Superdense coding.** Superdense coding [Bennett and Wiesner, 1992] is a quantum communication protocol that allows Alice to transmit two classical bits of information to Bob by sending only one qubit, given that they share a pair of entangled qubits. The protocol can be divided into five steps:

1. **Preparation:** Charlie prepares a maximally entangled Bell state, such as $|\beta_{00}\rangle = \frac{1}{\sqrt{2}}(|00\rangle + |11\rangle)$.

2. **Sharing:** Charlie sends the qubit 1 to Alice and the qubit 2 to Bob. Alice and Bob can be separated by an arbitrary distance.

3. **Encoding:** Depending on the two classical bits $zx \in \{00, 01, 10, 11\}$ that Alice wants to send, she applies the corresponding quantum gate operation to her qubit, transforming the

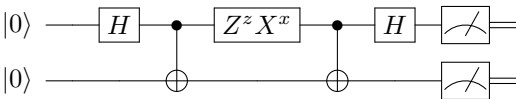

Figure 5: Quantum circuit diagram for superdense coding.

Bell state $|\beta_{00}\rangle$ into one of the four Bell states:

$$|\beta_{00}\rangle = \frac{1}{\sqrt{2}}(|00\rangle + |11\rangle) \text{ if } zx = 00$$

$$|\beta_{01}\rangle = \frac{1}{\sqrt{2}}(|01\rangle + |10\rangle) \text{ if } zx = 01$$

$$|\beta_{10}\rangle = \frac{1}{\sqrt{2}}(|00\rangle - |11\rangle) \text{ if } zx = 10$$

$$|\beta_{11}\rangle = \frac{1}{\sqrt{2}}(|01\rangle - |10\rangle) \text{ if } zx = 11$$

Alice achieves these transformations by applying the operation $Z^z X^x$ to her qubit, where $Z$ is the phase-flip gate, $X$ is the bit-flip gate. Specifically:

- If $zx = 00$, Alice applies $Z^0 X^0 = I$ (identity gate).
- If $zx = 01$, Alice applies $Z^0 X^1 = X$ (bit-flip gate).
- If $zx = 10$, Alice applies $Z^1 X^0 = Z$ (phase-flip gate).
- If $zx = 11$, Alice applies $Z^1 X^1 = ZX = iY$ gate.

4. **Sending:** Alice sends her qubit to Bob through a quantum channel.

5. **Decoding:** Bob applies a CNOT gate followed by a Hadamard gate to the two qubits, transforming the entangled state into the corresponding computational basis state $|zx\rangle$. By measuring the qubits, Bob obtains the two classical bits $zx$ sent by Alice.

Superdense coding exploits the properties of quantum entanglement to transmit two classical bits of information using only one qubit. The quantum circuit diagram for superdense coding is shown in Figure 5.

**Quantum teleportation.** Quantum teleportation [Bennett et al., 1993] is a technique for transferring quantum information from a sender (Alice) to a receiver (Bob) using shared entanglement and classical communication. The protocol can be described as follows:

1. **Preparation:** Telamon prepares a maximally entangled Bell state, such as $|\beta_{00}\rangle = \frac{1}{\sqrt{2}}(|00\rangle + |11\rangle)$.

2. **Sharing:** Alice has qubit 1 in the state $|\psi\rangle = \alpha|0\rangle + \beta|1\rangle$, which she wants to teleport to Bob. Telamon shares qubit 2 with Alice and qubit 3 with Bob, creating the shared entangled state $|\beta_{00}\rangle_{23}$.

3. **Encoding:** Alice wants to teleport an unknown quantum state $|\psi\rangle = \alpha|0\rangle + \beta|1\rangle$ to Bob. She applies a CNOT gate to qubits 1 and 2, with qubit 1 as the control and qubit 2 as the target. Then, she applies a Hadamard gate to qubit 1. The resulting state of the three-qubit system is:

$$|\Psi\rangle = \frac{1}{2}[|\beta_{00}\rangle(\alpha|0\rangle + \beta|1\rangle) + |\beta_{01}\rangle(\alpha|1\rangle + \beta|0\rangle)$$
$$+ |\beta_{10}\rangle(\alpha|0\rangle - \beta|1\rangle) + |\beta_{11}\rangle(\alpha|1\rangle - \beta|0\rangle)].$$

4. **Measurement:** Alice measures qubits 1 and 2 in the Bell basis and obtains one of four possible outcomes: $|\beta_{00}\rangle$, $|\beta_{01}\rangle$, $|\beta_{10}\rangle$, or $|\beta_{11}\rangle$. This measurement collapses the three-qubit

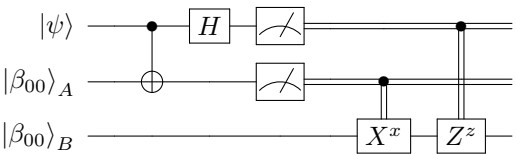

Figure 6: Quantum circuit diagram for quantum teleportation

state into one of the following:

$$|\beta_{00}\rangle \otimes (\alpha|0\rangle + \beta|1\rangle)$$
$$|\beta_{01}\rangle \otimes (\alpha|1\rangle + \beta|0\rangle)$$
$$|\beta_{10}\rangle \otimes (\alpha|0\rangle - \beta|1\rangle)$$
$$|\beta_{11}\rangle \otimes (\alpha|1\rangle - \beta|0\rangle)$$

5. **Classical Communication:** Alice sends the result of her measurement (two classical bits) to Bob via a classical channel.

6. **Reconstruction:** Depending on the classical information received from Alice, Bob applies the operation $Z^z X^x$ to qubit 3, where $z$ and $x$ correspond to the two classical bits sent by Alice:

   - If Alice measured $|\beta_{00}\rangle$, she sends $zx = 00$, and Bob applies $Z^0 X^0 = I$ (identity operation).
   - If Alice measured $|\beta_{01}\rangle$, she sends $zx = 01$, and Bob applies $Z^0 X^1 = X$ (bit-flip).
   - If Alice measured $|\beta_{10}\rangle$, she sends $zx = 10$, and Bob applies $Z^1 X^0 = Z$ (phase-flip).
   - If Alice measured $|\beta_{11}\rangle$, she sends $zx = 11$, and Bob applies $Z^1 X^1 = ZX = iY$ (bit-flip and phase-flip).

   After applying the appropriate operation, Bob's qubit 3 will be in the state $|\psi\rangle = \alpha|0\rangle + \beta|1\rangle$, which is the original state that Alice wanted to teleport.

The quantum circuit diagram for quantum teleportation is shown in Figure 6.

**Quantum key distribution.** Quantum key distribution (QKD) [Bennett and Brassard, 1984] is a secure communication protocol that allows two parties, Alice and Bob, to produce a shared random secret key, which can then be used to encrypt and decrypt messages. The security of QKD is based on the fundamental principles of quantum mechanics that measuring a qubit can change its state. One of the most well-known QKD protocols is the BB84 protocol, which works as follows:

1. Alice randomly generates a bit string and chooses a random basis (X or Z) for each bit. She then encodes the bits into qubits using the chosen bases and sends them to Bob through a quantum channel.

2. Bob measures the received qubits in randomly chosen bases (X or Z) and records the results.

3. Alice and Bob communicate over a public classical channel to compare their basis choices. They keep only the bits for which their basis choices coincide and discard the rest.

4. Alice and Bob randomly select a subset of the remaining bits and compare their values. If the error rate is below a certain threshold, they conclude that no eavesdropping has occurred, and the remaining bits can be used as a secret key. If the error rate is too high, they abort the protocol, as it indicates the presence of an eavesdropper (Eve).

The security of the BB84 protocol relies on the fact that any attempt by Eve to measure the qubits during transmission will introduce detectable errors, alerting Alice and Bob to the presence of an eavesdropper.

# C  Additional Experiment Results

## C.1  Metrics

**BLEU Score.**   Bilingual Evaluation Understudy (BLEU) score is a metric used to evaluate the quality of machine-translated text compared to human-translated text. It measures how close the machine translation is to one or more reference translations. The BLEU score evaluates the quality of text generated by comparing it with one or more reference texts. It does this by calculating the n-gram precision, which means it looks at the overlap of n-grams (contiguous sequences of n words) between the generated text and the reference text. Originally the BLEU score ranges from 0 to 1, where 1 indicates a perfect match with the reference translations. Here rescaling the score makes it ranges from 0 to 100.

The BLEU score, originally designed for machine translation, can also be effectively used for evaluating algorithm generation tasks. Just as BLEU measures the similarity between machine-translated text and human reference translations, it can measure the similarity between a generated algorithm and a gold-standard algorithm. This involves comparing sequences of tokens to assess how closely the generated output matches the reference solution. In the context of algorithm generation, n-grams can represent sequences of tokens or operations in the code. BLEU score captures the precision of these n-grams, ensuring that the generated code aligns closely with the expected sequences found in the reference implementation.

The formula for BLEU score is given by:

$$\text{BLEU} = BP \cdot \exp\left(\sum_{n=1}^{N} w_n \log p_n\right).$$

where $BP$ is the acronym for brevity penalty, $w_n$ is the weight for the n-gram precision (typically $\frac{1}{N}$ for uniform weights), $p_n$ is the precision for n-grams. BP is calculated as:

$$BP = \begin{cases} 1 & \text{if } c > r \\ e^{1-\frac{r}{c}} & \text{if } c \leq r \end{cases}.$$

where $c$ is the length of the generated text and $r$ is the length of the reference text. Furthermore, n-gram precision $p_n$ is calculated as:

$$p_n = \frac{\sum_{C \in \text{Candidates}} \sum_{n-\text{gram} \in C} \min(\text{Count}(n-\text{gram in candidate}), \text{Count}(n-\text{gram in references}))}{\sum_{C \in \text{Candidates}} \sum_{n-\text{gram} \in C} \text{Count}(n-\text{gram in candidate})}.$$

This formulation ensures that the BLEU score takes into account both the precision of the generated n-grams and the overall length of the translation, providing a balanced evaluation metric.

**Byte Perplexity.**   Perplexity is a measure of how well a probability distribution or a probabilistic model predicts a sample. In the context of language models, it quantifies the uncertainty of the model when it comes to predicting the next element in a sequence. Byte perplexity specifically deals with sequences of bytes, which are the raw binary data units used in computer systems. For our purposes, we consider byte perplexity under UTF-8 encoding, a widely used character encoding standard that represents each character as one or more bytes.

For a given language model, let $p(x_i|x_{<i})$ be the probability of the $i$-th byte $x_i$ given the preceding bytes $x_{<i}$. If we have a sequence of bytes $x = (x_1, x_2, \ldots, x_N)$, the perplexity $PP(x)$ of the model on this sequence is defined as:

$$PP(x) = 2^{-\frac{1}{N} \sum_{i=1}^{N} \log_2 p(x_i|x_{<i})}.$$

A notable feature of byte perplexity is that, it does not rely on any specific tokenizer, making it versatile for comparing different models. Therefore, byte perplexity can be used to measure the

performance in quantum algorithm generation tasks. In such tasks, a lower byte perplexity indicates a better-performing model, as it means the model is more confident in its predictions of the next byte in the sequence.

## C.2 Experiment Results

Due to considerable variances in the experiments, we conducted additional rounds to obtain more representative data.

The BLEU scores for various quantum algorithm design tasks are illustrated in Figure 7. This figure not only displays the average performance of each model but also highlights the differences in performance across individual quantum algorithm tasks. The first notable observation is that the figure clearly demonstrates the varying levels of difficulty among quantum algorithms. For example, models achieve higher BLEU scores on tasks such as Bernstein-Vazirani and Deutsch-Jozsa, whereas they perform significantly worse on tasks like Grover, phase estimation, and quantum Fourier transform. This indicates that the former tasks are considerably easier than the latter ones. Another significant observation is that most models score higher in a five-shot prompt compared to a one-shot prompt, which confirms the large language models' ability to improve performance through contextual learning.

Similar patterns are observed in oracle construction tasks, as illustrated in Figure 8. The figure highlights that the Diffusion Operator task is notably more challenging than the Grover oracle construction task. Interestingly, we found that adding more in-context examples actually reduced the performance of the Phi-3-medium-128k-instruct and Mistral-7B-v0.3 models. This decline in performance could be attributed to the significant differences between each oracle construction task, which may be too out-of-distribution. Consequently, the additional examples might cause the models to overfit to the specific examples provided in the context, rather than generalizing well across different tasks.

In addition to evaluating the BLEU score, we conducted an experiment to measure the correctness of the machine-generated algorithms, and the results are shown in Table 3.[3] By running a verification function, we discovered that phase estimation and the swap test are significantly more challenging than other problems, leading most models to score -1 (indicating they cannot even generate the correct syntax). Notably, the BLEU score for the swap test is above average compared to other algorithms, yet almost none of the models produced a correct algorithm. This discrepancy highlights a critical limitation of using BLEU as a metric for algorithm evaluation. BLEU measures average similarity, but even a single mistake in an algorithm can render it entirely incorrect, thus failing to capture the true accuracy and functionality of the generated algorithms. Another important finding is that in a five-shot setting, GPT-4 and GPT-3.5 surpass all other models by a large margin. This demonstrates their exceptional capabilities, particularly in long-context comprehension and in-context learning. These models not only excel in understanding and generating text based on minimal examples but also maintain high performance over extended sequences, highlighting their advanced architecture and training methodologies.

---

[3]When we prepare supplementary materials, we observe that these experiments have considerable variances, and they are hence executed by additional rounds to obtain more representative data. As a result, we refreshed the data in Table 1 and Table 2 in the main body, and we also place Table 3 and Table 4 here.

Table 3: Benchmarking algorithm design in verification function scores.

| Model | Shot | Bernstein-Vazirani | Deutsch-Jozsa | Grover | Phase Estimation | Quantum Fourier Transform | Simon | GHZ State | Random Number Generator | Swap Test | W State | Average |
|---|---|---|---|---|---|---|---|---|---|---|---|---|
| gpt-4o-2024-05-13 | 1 | -0.8462 | -0.5538 | -0.7089 | -1.0000 | -1.0000 | -0.6692 | -0.8462 | -1.0000 | -1.0000 | -1.0000 | -0.8624 |
| gpt-4o-2024-05-13 | 5 | -0.3054 | 0.0135 | -0.2071 | -0.5357 | -0.6154 | -0.3692 | -0.1538 | -0.4967 | -0.8700 | -0.9231 | -0.4463 |
| Meta-Llama-3-8B | 1 | -0.2308 | -0.7692 | -0.7143 | -0.8571 | -0.9231 | -1.0000 | -0.6154 | -0.9285 | -1.0000 | -0.3846 | -0.7423 |
| Meta-Llama-3-8B | 5 | 0.0769 | -0.2308 | -0.5393 | -1.0000 | -0.7692 | -0.8462 | -0.3846 | -0.7276 | -1.0000 | -0.1538 | -0.5575 |
| gpt-3.5-turbo-0125 | 1 | -0.8462 | -0.7154 | -0.5679 | -1.0000 | -1.0000 | -0.6231 | -0.8462 | -1.0000 | -1.0000 | -1.0000 | -0.8599 |
| gpt-3.5-turbo-0125 | 5 | -0.6154 | -0.0571 | -0.0500 | -1.0000 | -0.6538 | -0.1646 | -0.2308 | -0.4513 | -0.8778 | -0.8462 | -0.4947 |
| Phi-3-medium-128k-instruct | 1 | -0.8462 | -0.7750 | -1.0000 | -1.0000 | -1.0000 | -1.0000 | -0.3846 | -1.0000 | -0.8878 | -0.8462 | -0.8740 |
| Phi-3-medium-128k-instruct | 5 | -0.6577 | -0.3821 | -0.8286 | -1.0000 | -1.0000 | -0.6100 | -0.9231 | -0.3569 | -0.8333 | -0.8462 | -0.7438 |
| Mistral-7B-v0.3 | 1 | -0.8462 | -0.8590 | -0.7107 | -1.0000 | -1.0000 | -0.9192 | -0.7692 | -1.0000 | -1.0000 | -0.6923 | -0.8797 |
| Mistral-7B-v0.3 | 5 | -0.6246 | -0.6667 | -0.4071 | -0.8571 | -0.9231 | -0.9115 | -0.6923 | -0.8820 | -1.0000 | -0.5385 | -0.7503 |

The verification results of the oracle construction task, as shown in Table 4, confirm our previous conclusions. In the five-shot setting, GPT-4 and GPT-3.5 consistently outperform all other models. Additionally, this table highlights the inconsistency between BLEU scores and verification scores. For instance, while the Diffusion Operator task achieves the lowest BLEU score, it is the Grover oracle construction that receives the lowest verification score. This discrepancy suggests that BLEU scores may not fully capture the performance of models in certain complex tasks.

Table 4: Benchmarking oracle construction in verification function scores.

| Model | Shot | Bernstein-Vazirani | Deutsch-Jozsa | Diffusion-Operator | Grover | Simon | Average |
|---|---|---|---|---|---|---|---|
| gpt-4o-2024-05-13 | 1 | -0.3200 | -0.0100 | -0.8462 | -0.9885 | -0.4674 | -0.5264 |
| gpt-4o-2024-05-13 | 5 | -0.1100 | 0.0800 | -0.3077 | -0.9540 | -0.0870 | -0.2757 |
| Meta-Llama-3-8B | 1 | -0.7300 | -0.5000 | -0.3846 | -1.0000 | -0.6848 | -0.6599 |
| Meta-Llama-3-8B | 5 | -0.0500 | 0.1700 | -0.8462 | -1.0000 | -0.6413 | -0.4735 |
| gpt-3.5-turbo-0125 | 1 | -0.3500 | -0.0400 | -0.8462 | -1.0000 | -0.3696 | -0.5211 |
| gpt-3.5-turbo-0125 | 5 | -0.1100 | 0.0200 | -0.3077 | -0.9770 | -0.1087 | -0.2967 |
| Phi-3-medium-128k-instruct | 1 | -0.6800 | -0.6100 | -0.9231 | -1.0000 | -0.7500 | -0.7926 |
| Phi-3-medium-128k-instruct | 5 | -0.5400 | -0.4300 | -1.0000 | -1.0000 | -0.8370 | -0.7614 |
| Mistral-7B-v0.3 | 1 | -0.4000 | -0.4300 | -0.9231 | -0.9540 | -0.6087 | -0.6632 |
| Mistral-7B-v0.3 | 5 | -0.3700 | -0.1300 | -1.0000 | -0.9195 | -0.2391 | -0.5317 |

The Byte Perplexity results, shown in Figure 9 and Figure 10, provide valuable insights into the performance of our model. Evaluated in a zero-shot setting, byte perplexity trends closely mirror those observed with BLEU scores. This alignment suggests that our model's predictive capabilities are consistent across Perplexity and BLEU evaluation metrics. Specifically, in the context of quantum algorithm design tasks, the results indicate that the Bernstein-Vazirani and Deutsch-Jozsa algorithms are relatively straightforward for the model, whereas the Simon algorithm presents greater difficulty. This differentiation highlights the varying levels of complexity inherent in these quantum algorithms.

### C.3 Case Study

After carefully examining the model's output, we observed several interesting patterns. We present a series of case studies to illustrate these observations and provide possible explanations.

**Low Score for GPT-4o in One-Shot Setting.** At first glance, it is surprising that GPT-4o performs poorly on many quantum algorithms in the algorithm design task in the one-shot setting compared to Llama3-8B. Given that Llama3-8B has a relatively smaller parameter scale, the results should have been the other way around. A closer examination of the model's output reveals the potential reason: while Llama3-8B closely mimics the input examples, GPT-4o tends to improvise, resulting in outputs that are not well captured by the current syntax support. Here are several concrete examples.

This is the OpenQASM 3.0 code output for the W state with $n = 7$. In this code, GPT-4o uses the advanced "for" loop syntax newly introduced in OpenQASM 3.0 to create the circuit. Although the code fails to produce the W state, it is syntactically correct. However, the Qiskit.qasm3 import module, which converts OpenQASM 3.0 files to QuantumCircuit objects and is used in our verification

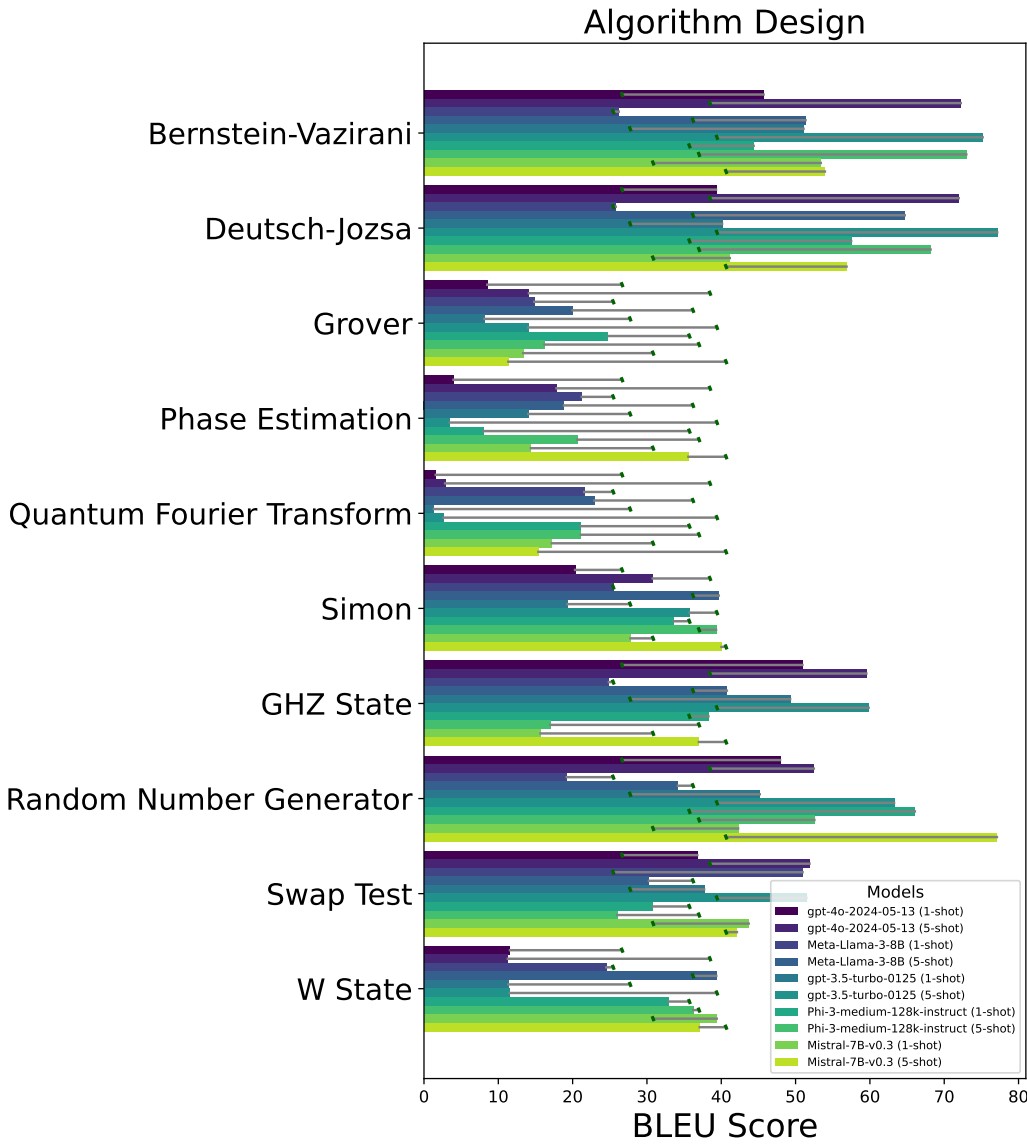

Figure 7: Benchmarking algorithm design in BLEU scores. The green dots represent each model's mean BLEU score across 10 algorithms, while the gray lines show how much its score on each algorithm deviates from this mean.

function to check the correctness of the syntax of output OpenQASM codes, is still in the experimental stage and does not support many of OpenQASM 3.0's advanced features. As a result, GPT-4o's use of these features causes the code to fail syntax validation, resulting in a score of -1.

```
OPENQASM 3.0;
include "stdgates.inc";
qubit[7] q;
h q[0];
for i in[1:6] {
    cx q[i-1], q[i];
}
```

Listing 8: OpenQASM 3.0 Code output by GPT-4o for W state with $n = 7$.

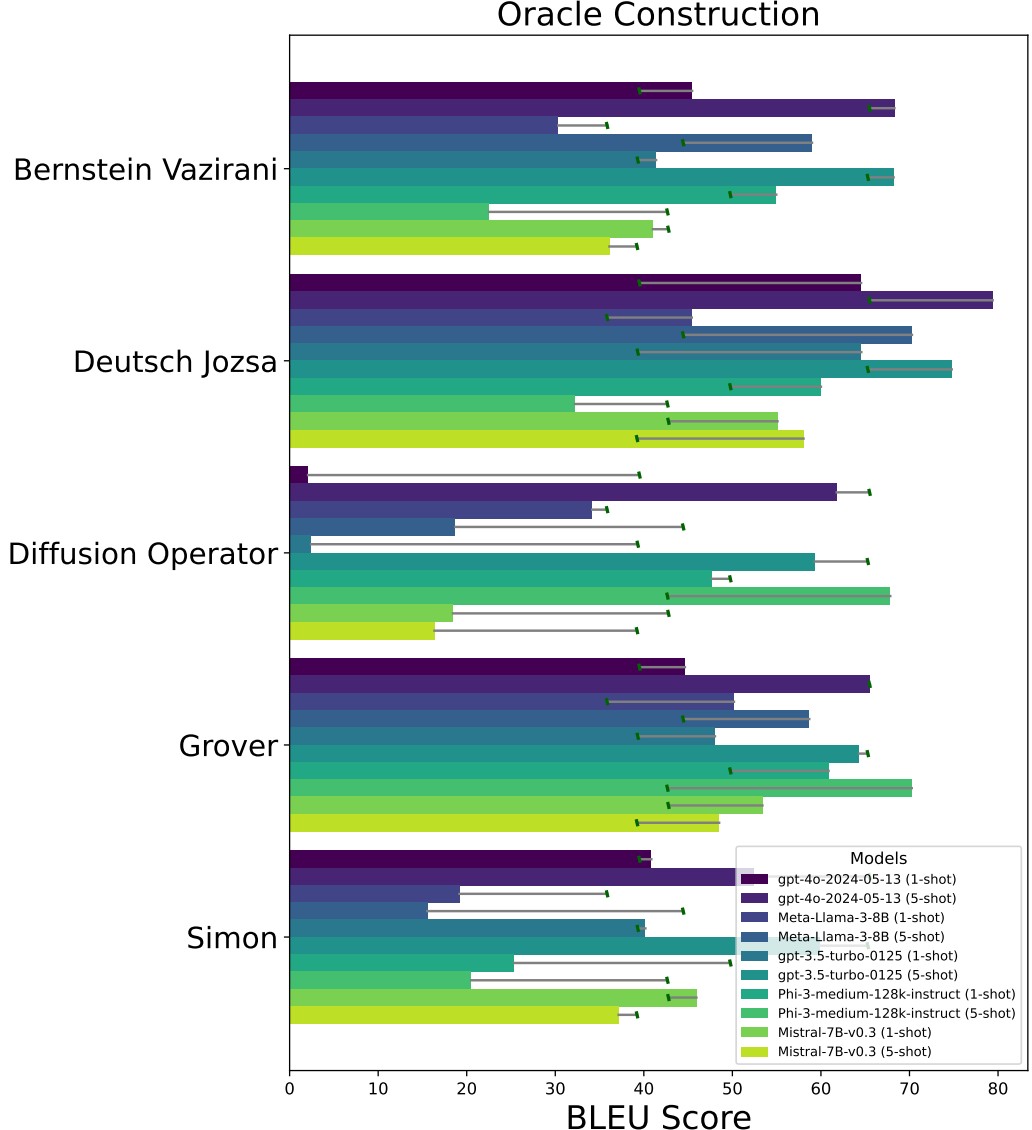

Figure 8: Benchmarking oracle construction in BLEU scores. The green dots represent each model's mean BLEU score across 5 oracles, while the gray lines show how much its score on each oracle deviates from this mean

Here is another example where GPT-4o decides to assign novel names to its qubit registers, leading to a conflict in the symbol table in Scope.GLOBAL. If we substitute all the registers $x$, $y$, and $s$ with new names, the code can pass syntax validation successfully and is close to the correct solution.

```
OPENQASM 3.0;
include "stdgates.inc";
include "oracle.inc";
bit[9] s;
qubit[10] x;
qubit[11] y;
h x[0];
h x[1];
h x[2];
h x[3];
```

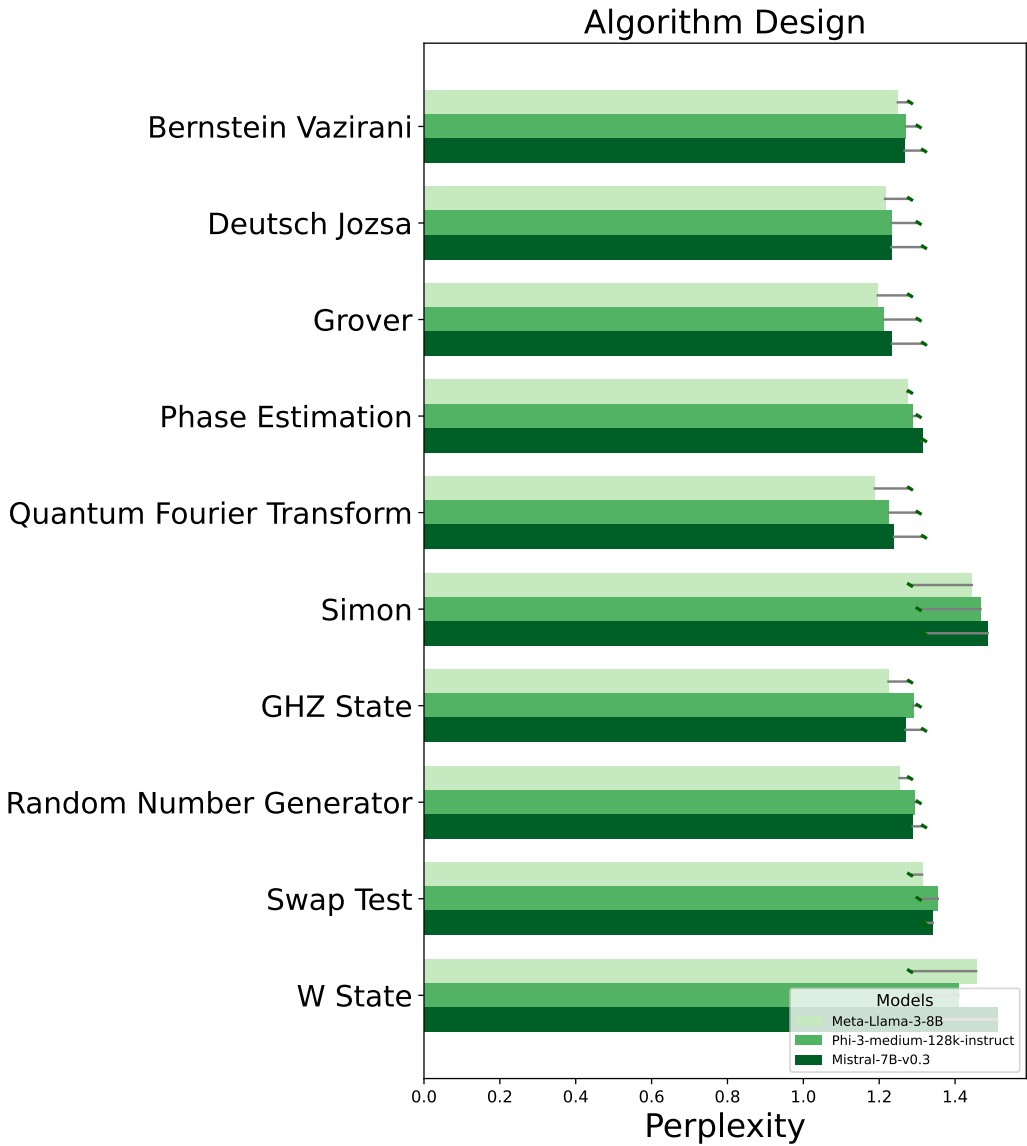

Figure 9: Benchmarking algorithm design in perplexity. The green dots represent each model's mean perplexity score across 10 algorithms, while the gray lines show how much its score on each algorithm deviates from this mean.

```
h x[4];
h x[5];
h x[6];
h x[7];
h x[8];
Oracle x[0], x[1], x[2], x[3], x[4], x[5], x[6], x[7], x[8], y;
h x[0];
h x[1];
h x[2];
h x[3];
h x[4];
h x[5];
h x[6];
h x[7];
```

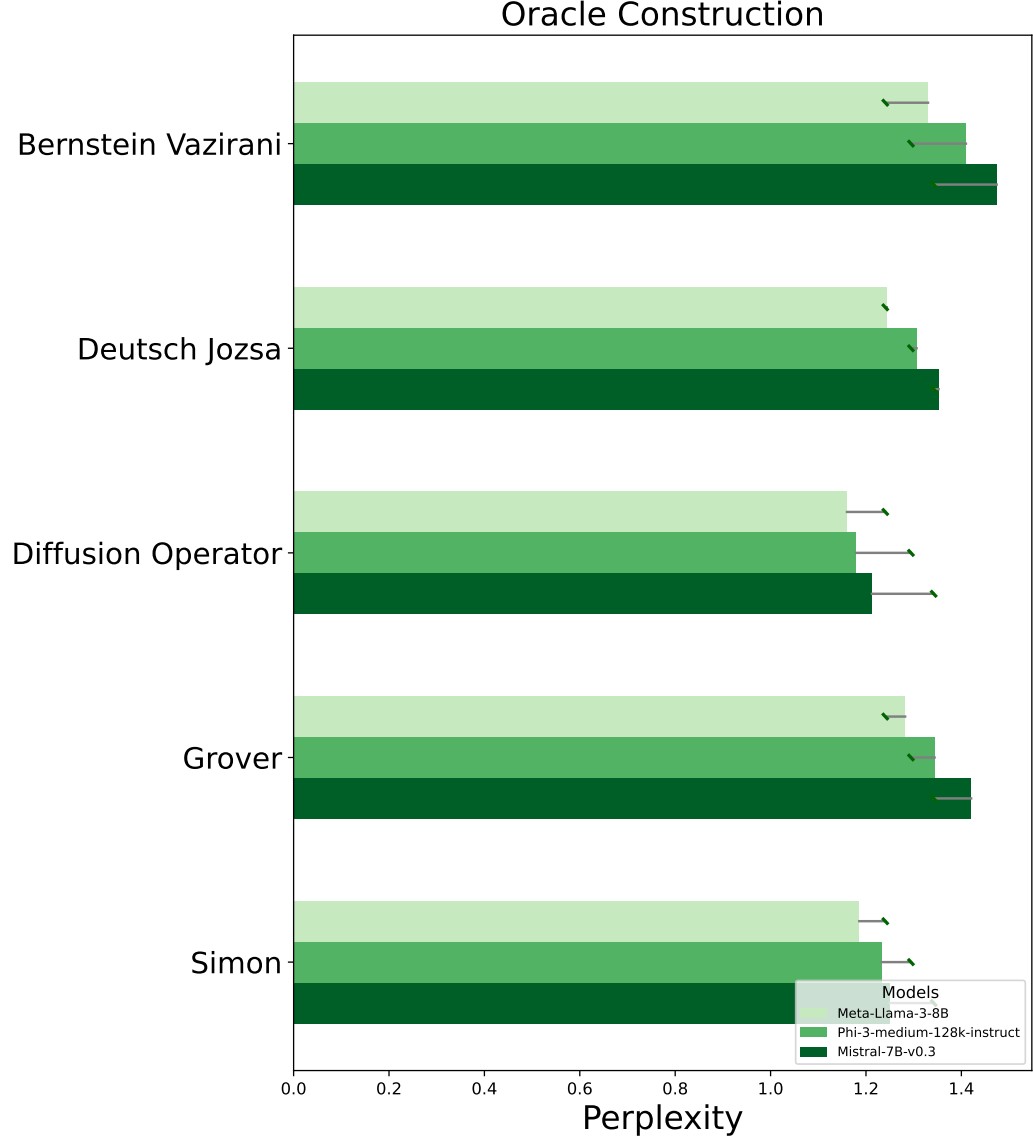

Figure 10: Benchmarking oracle construction in perplexity. The green dots represent each model's mean perplexity score across 5 oracles, while the gray lines show how much its score on each oracle deviates from this mean.

```
h x[8];
measure x[0] -> s[0];
measure x[1] -> s[1];
measure x[2] -> s[2];
measure x[3] -> s[3];
measure x[4] -> s[4];
measure x[5] -> s[5];
measure x[6] -> s[6];
measure x[7] -> s[7];
measure x[8] -> s[8];
```

Listing 9: OpenQASM 3.0 Code output by GPT-4o for Bernstein-Vazirani Problem with $n = 9$.

Since Llama3-8B tends to follow the provided example more closely, it avoids errors like this. These factors may help explain why GPT-4o performs poorly in the one-shot setting compared to Llama3-8B.

**High Score for GPT-4o in Five-Shot Setting.** In contrast to its relatively low score in the one-shot setting, GPT-4o achieves the best overall performance in the five-shot setting for both oracle construction and algorithm design tasks.

Unlike in the one-shot setting, where GPT-4o frequently uses advanced OpenQASM 3.0 features, it learns from the examples in context and uses simpler syntax in the five-shot setting. For the W state task, here is an output in the five-shot setting:

```
OPENQASM 3.0;
include "stdgates.inc";
qubit[3] q;
h q[0];
cx q[0], q[1];
h q[1];
cx q[1], q[2];
h q[2];
```

Listing 10: OpenQASM 3.0 Code output by GPT-4o for W state with $n = 3$.

This output avoids the use of "for" loops and successfully passes the syntax validation test, although it still cannot generate the W state correctly.

In addition to adapting to plain syntax through in-context learning, GPT-4o achieves outstanding performance on more complicated tasks such as phase estimation. Here is the model output of GPT-4o on the phase estimation task with qubit number $n = 2$.

```
OPENQASM 3.0;
include "stdgates.inc";
include "oracle.inc";
bit[2] c;
qubit[2] q;
Psi q[0];
h q[1];
CU_0 q[0], q[1];
h q[1];
c[0] = measure q[1];
c[1] = measure q[0];
```

Listing 11: OpenQASM 3.0 Code output by GPT-4o for Phase Estimation with $n = 2$.

```
from qiskit import transpile

def run_and_analyze(circuit, aer_sim):
    """Run the circuit and analyze the result."""
    circ = transpile(circuit, aer_sim)
    result = aer_sim.run(circ, shots=1).result()
    counts = result.get_counts()
    phase_str = list(counts.keys())[0]
    phase = int(phase_str, 2) / 2**2
    return phase
```

Listing 12: Post-processing code output by GPT-4o for Phase Estimation with $n = 2$.

This suite of OpenQASM 3.0 circuits and post-processing functions successfully outputs the phase within the required precision for the test case, resulting in an impressive verification score of 1.0. Despite the small number of qubits and differences from the reference implementation, the accuracy achieved is noteworthy.

These phenomena reflect that GPT-4o has impressive in-context learning abilities and overall better capabilities in designing and implementing quantum algorithms.