# OpenReview forum: "QCircuitNet: A Large-Scale Hierarchical Dataset for Quantum Algorithm Design"
_NeurIPS.cc/2024/Datasets_and_Benchmarks_Track — Submitted to NeurIPS 2024 Track Datasets and Benchmarks_

### Official Review · Reviewer_DaGg · 2024-07-07
**Interesting first steps for an important problem**

**Rating:** 7
**Confidence:** 4
**Correctness:** The claims and methodology in the pap…

**Review:**

The authors approach the problem of quantum algorithm design by regarding quantum circuits as programming languages, and based on this, designed and produced the dataset. This change of perspective is important and allows the dataset to be scalable to large system sizes. This makes it useful for the benchmarking and evaluation of ML methods for quantum algorithm design. However, the current size of the dataset is still too small (in terms of the number of different quantum algorithms) to be useful, which nevertheless is tolerable for preliminary works on this direction.

**Strengths:**

1. The perspective change of treating quantum algorithms as languages is interesting and fruitful.
2. The dataset encompasses the important features of quantum algorithm designs: problem setting, circuit description, post-processing, verification. The authors also take good care of how to properly encode them.
3. The dataset contains most of the established and important quantum algorithms/primitives.

**Additional Feedback:**

None.

**Clarity:**

The paper is clearly written up to the following minor typos.
1. line 27, "the design quantum algorithms" -> "the design of quantum algorithms"
2. line 38, "As far we know" -> "As far as we know"

**Documentation:**

Good ducumentation.

**Ethics:**

Not applicable.

**Limitations:**

1. The authors should remove the "scalable" and "efficient" description of their evaluation methods. For example, "allowing for scalable and efficient evaluation." in line 48. This is because the evaluation of the quantum circuits requires simulating the exponentially large state evolution, say in qiskit, which in general needs exponential time (i.e., not scalable and inefficient). On the other hand, the more crude metrics like BLEU or perplexity can indeed be efficiently evaluated, since they are directly calculated based on the circuit descriptions.
2. Experiments that train on this dataset from scratch are not carried out. I believe this would be significantly more interesting than the LLMs which are never targeted at this specific tasks in the first place. This however is clearly stated by the authors as a direction for future works.
3. Directly targeting at the problem of quantum algorithm design might be too ambitious. This is also evident from the poor performance reported. It would be more interesting to look at easier tasks based on the circuit descriptions.

**Opportunities For Improvement:**

1. The authors could emphasize more on the theoretical side of the benifits to use circuit description as the encoding for quantum algorithms. In particular, the full description of quantum states and unitary gates are exponentially costly, but the meaningful quantum algorithms often only consists of at most $\log n$ or less gates (e.g., Shor's algorithm). And in general, those that can be implemented by Nature efficiently must not have more than polynomially many gates (see e.g.,
https://doi.org/10.1103/PhysRevLett.106.170501). So, theoretically speaking, using circuit descriptions can indeed allow scalable representations. I believe that the authors should emphasize more of this point.
2. The authors should think over how to deal with the small size of the dataset. The number of useful quantum algorithms known is quite small, and this also makes the dataset quite small in terms of variety of algorithms.Turning to random quantum/Clifford circuits might be a way out. The corresponding problem could be to take the zero states to some state generated by a certain circuit. This could significantly enlagre the size of the dataset.
3. The authors should report the standard deviation of the scores in Table 1, 2, etc.

**Relation To Prior Work:**

Relation to prior work is properly stated.

**Summary And Contributions:**

This work introduces QCircuitNet, a dataset containing code descriptions of various important quantum algorithms. Alongside the code, natural language descriptions of the corresponding computation problems, post-processing functions, and verification methods are also provided. The objective is to facilitate the training of machine learning algorithm for designing (ideally new) quantum algorithms. The dataset is further fed into several established large language models (LLMs) as few-shot training prompts to test the performance of these LLMs. Experimental results show that the designated task is far beyond the current capability of these LLMs.

---

> ### Author Rebuttal · Authors · 2024-08-17
>
> **(1) The authors could emphasize more on the theoretical side of the benifits to use circuit description as the encoding for quantum algorithms. In particular, the full description of quantum states and unitary gates are exponentially costly, but the meaningful quantum algorithms often only consists of at most or less gates (e.g., Shor's algorithm). And in general, those that can be implemented by Nature efficiently must not have more than polynomially many gates (see e.g., https://doi.org/10.1103/PhysRevLett.106.170501). So, theoretically speaking, using circuit descriptions can indeed allow scalable representations. I believe that the authors should emphasize more of this point.**
>
> We sincerely thank the reviewer for this valuable insight. We acknowledge that we previously overlooked this important theoretical point. In the final version, we will add a detailed discussion to emphasize the theoretical benefits of using circuit descriptions for encoding quantum algorithms.
>
>
>
> **(2) The authors should think over how to deal with the small size of the dataset. The number of useful quantum algorithms known is quite small, and this also makes the dataset quite small in terms of variety of algorithms. Turning to random quantum/Clifford circuits might be a way out. The corresponding problem could be to take the zero states to some state generated by a certain circuit. This could significantly enlagre the size of the dataset.**
>
> We totally agree with the reviewer that the limited size of existing quantum algorithms is the main bottleneck restricting the size of the dataset in terms of variety of quantum algorithms. We have implemented random circuits as suggested to enlarge the dataset. To represent different levels of difficulties for generation and simulation, we implemented circuits with Clifford gate set \{H, S, CNOT\} and non-Clifford set \{H, S, T, CNOT\} separately.
>
> In addition, we would like to mention that the Quantum Logic Synthesis part of the Oracle Construction task aligns with this purpose as well, since there are exponentially many secret strings or mapping functions to choose for the oracle, resulting in significantly large data amount. Inspired by this observation, we combined data from the oracle construction task with newly generated random circuits, and created a dataset comprising quantum circuits tailored for specific transformations with 227067 entries (easily expandable). We further conducted few-shot learning \& fine-tuning experiments on this dataset. The results are provided in General Response (4).
>
> **(3) The authors should report the standard deviation of the scores in Table 1, 2, etc.**
>
> We thank the reviewer for the suggestion. The standard errors (SE) of the experiments are included in the attached PDF, as SE is usually the default metric in such experiments. The relationship between standard error (SE) and standard deviation (SD) is given by $\text{SE} = \frac{\text{SD}}{\sqrt{n}}$, where $n$ is the number of data points in each experiment.

---

> ### Author Rebuttal · Authors · 2024-08-17
>
> **(4) The authors should remove the "scalable" and "efficient" description of their evaluation methods. For example, "allowing for scalable and efficient evaluation." in line 48. This is because the evaluation of the quantum circuits requires simulating the exponentially large state evolution, say in qiskit, which in general needs exponential time (i.e., not scalable and inefficient). On the other hand, the more crude metrics like BLEU or perplexity can indeed be efficiently evaluated, since they are directly calculated based on the circuit descriptions.**
>
> We appreciate the reviewer's suggestion and will remove the "scalable" and "efficient" description in the revised paper. Indeed classical simulation of quantum circuits is costly and not scalable. We initially adopted these words with the intention to express that directly verifying the results of quantum circuits could be more realistic and inclusive than comprehensive circuit logic equivalence checking, and metrics like BLEU and perplexity helped with efficient evaluation by treating circuits as languages. We apologize for any confusion caused.
>
> **(5) Experiments that train on this dataset from scratch are not carried out. I believe this would be significantly more interesting than the LLMs which are never targeted at this specific tasks in the first place. This however is clearly stated by the authors as a direction for future works.**
>
> We appreciate the reviewer’s suggestion. We initially adopted LLMs because of their ability to integrate extensive domain knowledge and their strong foundation in programming tasks. Training a model from scratch based on our current dataset, while a very intriguing direction, might face challenges such as generating syntactically correct OpenQASM / python code, creating meaningful post-processing functions, and lacking essential background knowledge in quantum computing. These challenges highlight the need for novel model architectures and training methods, and we are keen to explore this compelling topic in future research. To address the concern, we have included preliminary fine-tuning results in our General Response (4). We hope these additions will clarify our rationale and highlight the potential for future research inspired by this work.
>
> **(6) Directly targeting at the problem of quantum algorithm design might be too ambitious. This is also evident from the poor performance reported. It would be more interesting to look at easier tasks based on the circuit descriptions.**
>
> We agree with the reviewer that directly targeting at quantum algorithm design might be a little too ambitious. This is also the reason why we included the Oracle Construction task in the paper, aiming at an easier task of creating quantum circuits meeting a specific requirement. Following the reviewer's suggestion, we have also included more data points of random circuits. More details could be referred to the individual Response (2). Nonetheless, quantum algorithm design remains a crucial problem, and with this work, we attempt to make a first step in this direction.
>
> **(7) Typos.**
>
> We thank the reviewer for pointing out the typos. We will fix them in the final version of our paper.

---

> ### Comment · Reviewer_DaGg · 2024-08-22
> **Response to Rebuttal**
>
> I'd like to thank the authors for putting the effort to carry out additional experiments and answer my questions. I believe that most of my concerns have been addressed. These changes have made the manuscript more suitable for acceptance.
>
> That said, there's still one thing that puzzles me. In General Response (4), the authors report that the performance on Clifford and Universal decreases after fine-tuning. I wonder if the authors have an understanding of why this counter-intuitive thing happens? What are the detailed design of the Clifford/Universal tasks?

---

> > ### Author Rebuttal · Authors · 2024-08-24
> >
> > We appreciate the reviewer’s insightful question. We also noticed this interesting phenomenon of performance decrease on Clifford/Universal tasks after fine-tuning.
> >
> > The Clifford/Universal tasks are designed as follows: The dataset was generated by randomly sampling gates from the specified gate set and their corresponding qubits, with varying total gate counts. We then simulated the circuits to obtain the final quantum states. In each task, the problem description provides the vector of the final state, and the model is required to generate quantum circuits that reproduce this state using the provided gates. The verification function compares the fidelity between the generated state and the target state.
> >
> > In the rebuttal, we fine-tuned the model using 1,800 samples each from the original four oracle construction tasks and the newly introduced random circuit tasks (Clifford/Universal), totaling 10,800 data points. Interestingly, the BLEU score for Clifford dropped from 13 to 10. After receiving the reviewer’s feedback, we conducted additional experiments and fine-tuned the model on 4,800 samples specifically for the Clifford task. Surprisingly, the BLEU score further decreased from 13 to 7. Upon closer inspection, we observed that the model more frequently generated outputs with infinite loops and increased monotony, often producing repetitive gate patterns and repeatedly cycling over the same qubit after fine-tuning. To investigate this, we further conducted experiments with different "temperature" parameters, which control the randomness of predictions. Typically, lower temperatures make the model more conservative, while higher temperatures flatten the distribution, increasing the likelihood of generating originally less probable sequences. The results are shown as below:
> >
> > *Table 1：Fine-Tuning Results on Clifford Task Across Different Temperature Settings*
> >
> > | Model            | Setting       | Temperature | BLEU      | Verification   |
> > |------------------|---------------|-------------|----------------|--------------------|
> > | Meta-Llama-3-8B  | few-shot(5)   | 0           | 13.3796(±0.9508) | -0.6582(±0.0360)    |
> > |                  |               | 0.2         | 12.5688(±0.8276) | -0.6526(±0.0372)    |
> > |                  |               | 1           | 53.0431(±3.8422) | -0.1914(±0.0361)    |
> > | Meta-Llama-3-8B  | finetune      | 0           | 7.6261(±0.3433)  | -0.8895(±0.0247)    |
> > |                  |               | 0.2         | 13.8714(±0.6536) | -0.7873(±0.0306)    |
> > |                  |               | 1           | 32.5241(±2.0548) | -0.2072(±0.0358)    |
> >
> > One possible explanation for this counter-intuitive result lies in the challenge of encoding quantum state vectors within a language model. In the problem description, the target quantum state is represented by a complex vector with four decimal places of precision, where the dimension scales as $2^n$ with the number of qubits $n$. It is a well-known fact that LLMs generally struggle with very long floating-point numbers, which might contribute to the observed performance decline.
> >
> > Another potential reason could be overfitting during fine-tuning, particularly for tasks that require high output diversity. The varying degrees of intrinsic difficulty and the amount of relevant pre-training knowledge across different tasks likely played a role. Oracle constructions are relatively simple for the model to learn. For example, in the Bernstein-Vazirani algorithm, the model only needs to apply a CNOT gate at positions corresponding to '1' bits. In contrast, the random circuits in the Clifford and Universal tasks involve more general and complex quantum state transformations, making them significantly more challenging. These tasks are also less common during pre-training, which could have hindered the model's ability to generalize without overfitting. This challenge is one of the reasons we initially considered a few-shot learning approach to be suitable.
> >
> > While these are plausible hypotheses, we acknowledge that further investigation is required to draw definitive conclusions. Nonetheless, we consider this an intriguing topic that warrants additional research. We are happy to adopt relevant discussions in the final version of our paper.

---

> > > ### Comment · Reviewer_DaGg · 2024-08-26
> > >
> > > Thanks for the explanation. Indeed this is an interesting direction for future works.

---

> > > > ### Author Response · Authors · 2024-08-26
> > > >
> > > > Thanks!

---

### Official Review · Reviewer_oAQd · 2024-07-20
**Dataset and benchmarking task of quantum algorithm design using large language models**

**Rating:** 4
**Confidence:** 3
**Correctness:** Yes

**Review:**

Unfortunately, the results do not look promising at all and showcase the pitfall of adopting LLM type technology to perform complex tasks of quantum circuit generation or predicting the syntax of quantum programming language. The target quantum algorithms being predicted also seem to be toy problems rather than any relevant practical application. It is unclear whether there is any value to such a dataset for future purposes since it is very limited to the task of using only LLMs for generating algorithms given that the collection of problems tackled is also restrictive. There are also challenges associated with the implementation of improper quantum algorithms on quantum simulators or actual hardware devices. A quantum algorithm that is even partially inaccurate predicted by LLMs and suffers from non-scalability would be unrealizable to any extent.

**Strengths:**

The framework design is well discussed and the evaluation carried out for the benchmarking is good.

**Additional Feedback:**

No

**Clarity:**

The introduction could be written better and improved significantly. The quality of the rest of the paper was decent.

**Documentation:**

Yes

**Limitations:**

Please see above.

**Opportunities For Improvement:**

The major limitation is hindering the creativity involved in quantum algorithm design (take the example of variational quantum eigensolver algorithm) by using LLMs. I do not see the point in making the quantum algorithm design a black-box approach since adopting a quantum computing approach is only justified for special applications where there is a clear advantage over classical computing.

**Relation To Prior Work:**

Yes

**Summary And Contributions:**

The authors present a study where large language models (LLMs) are utilized for quantum algorithm design and oracle construction. For this purpose, the work highlights the various design principles considered to develop such a framework. Several LLMs are benchmarked using evaluation metrics such as BLEU score, byte perplexity, and verification function.

---

> ### Author Rebuttal · Authors · 2024-08-17
>
> We appreciate the reviewer for the comment. Indeed, the performance of current LLMs on our dataset demonstrates significant opportunity for improvement. This, however, reveals the value of our dataset as a benchmark and presents great research potential in turn.
>
> Quantum algorithm design is a very important problem in the research of quantum computing. As the discipline evolves, we aim to explore more possibilities, especially in line with recent advances among "AI for Science". In this work, we try to take a first step by making a change of perspective through formulating quantum algorithms as languages. As mentioned by Reviewer DaGg, this novel formulation is not only theoretically feasible, but also enables us to leverage the burgeoning technology of artificial intelligence in language processing and sequential modeling.
>
> We adopted the LLM-based methodology because it represents the best practice of sequential modeling methods at current stage. As verified by numerous studies, one-shot reasoning task is extremely hard for AI models. LLMs have an edge over other models in that they possess the best pre-training knowledge, and provide human-friendly interfaces which support human-machine collaboration. This is also why the mathematical community is now actively exploring LLM-based methods for mathematical reasoning, such as proving theorems [1] and optimizing theoretical bounds [2]. We have reasons to believe that one day LLMs will also be applicable in quantum computing.
>
> To accommodate with LLMs, we provided a carefully designed framework encompassing the key features of quantum algorithm design, including problem setting, quantum circuit codes, classical post-processing, and verification functions. We also made great efforts to maintain the black-box nature of oracles, use the shot number to characterize query complexity, and properly deal with many other details, which is highly non-trivial. The resulting framework not only supports the characterization of more advanced quantum algorithms, but also bears the flexibility to extend to other quantum related tasks, such as oracle construction and quantum logic synthesis.
>
> To address the concern of toy problems, we implemented Generalized Simon's Problem, which is a generalized and more advanced version of the standard Simon's problem. We refer the reviewer to General Response (1) for more details. For scalability issues, please see the discussion in General Response (3).
>
> Finally, we will polish through the introduction section in the final version of our paper, following your suggestion.
>
> [1] K. Yang, A. Swope, A. Gu, R. Chalamala, P. Song, S. Yu, S. Godil, R. J. Prenger, and A. Anandkumar. Leandojo: Theorem proving with retrieval-augmented language models. Advances in Neural Information Processing Systems, 36, 2024.
>
> [2] B. Romera-Paredes, M. Barekatain, A. Novikov, M. Balog, M. P. Kumar, E. Dupont, F. J. Ruiz, J. S. Ellenberg, P. Wang, O. Fawzi, et al. Mathematical discoveries from program search with large language models. Nature, 625(7995):468–475, 2024.

---

> ### Author Response · Authors · 2024-08-26
>
> Dear Reviewer,
>
> Thank you once again for the time and effort you've devoted to reviewing our paper and considering our rebuttal. As we approach the end of the discussion period, we would greatly appreciate any further feedback you might have. Your insights are invaluable, and we are keen to address any additional concerns to improve our work.
>
> We look forward to your response.

---

### Official Review · Reviewer_7rUP · 2024-08-02

**Rating:** 6
**Confidence:** 3
**Correctness:** Provided in the review section above.
**Clarity:** Provided in the review section above.

**Review:**

Strengths:-
- QCircuitNet addresses a significant gap in the field by providing the first comprehensive dataset for quantum algorithm design, which is crucial for advancing AI in quantum computing.
- The dataset's organization is well-thought-out, separating oracle definitions from algorithm circuits to maintain the black-box nature of oracles in quantum algorithm design.
- The framework is designed to be easily extendable to more advanced quantum algorithms, making it future-proof.
-  The use of multiple metrics (BLEU, byte perplexity, verification function) provides a comprehensive assessment of LLM performance.
- The authors address challenges such as the need for explicit gate implementations in experiment platforms while maintaining the theoretical black-box nature of oracles.
- By using QASM files created from scratch, the dataset helps mitigate potential biases from existing tutorials and benchmarks.

Weaknesses :-
-  While the dataset covers fundamental quantum algorithms, it may not fully represent the complexity of more advanced quantum algorithms used in research.
-  The binary nature of the verification function (correct/incorrect) may not capture partial correctness or near-misses in algorithm implementation.
- The dataset focuses on circuit design and simulation but doesn't address the challenges of implementing these algorithms on actual quantum hardware with noise and errors.
- While the paper suggests the potential for fine-tuning LLMs, it doesn't provide concrete experiments or results in this direction.
- While the paper presents performance metrics, it doesn't delve deeply into the types of errors made by LLMs or patterns in their mistakes.
- The paper doesn't thoroughly address how the approach scales with increasing qubit numbers or more complex quantum algorithms.

**Strengths:**

Provided in the review section above.

**Additional Feedback:**

Provided in the review section above.

**Documentation:**

Provided in the review section above.

**Ethics:**

Provided in the review section above.

**Limitations:**

Provided in the review section above.

**Opportunities For Improvement:**

Provided in the review section above.

**Relation To Prior Work:**

Provided in the review section above.

**Summary And Contributions:**

This paper introduces QCircuitNet, a novel dataset and benchmark for evaluating AI models' capabilities in quantum algorithm design and implementation. The key contributions include:

- A structured dataset for quantum algorithm design tasks, covering oracle construction and algorithm implementation.
- A framework for formulating quantum algorithm design tasks for Large Language Models (LLMs).
- Automatic validation and verification functions for efficient evaluation.
- A benchmark methodology to assess LLMs' performance on quantum computing tasks.

The dataset structure includes problem descriptions, generation code, algorithm circuits in OpenQASM 3.0 format, post-processing functions, oracle/gate definitions, verification functions, and dataset creation scripts. The authors benchmark several leading LLMs (GPT-3.5, GPT-4, LLAMA-3, Phi-3, and Mistral-7B) using metrics such as BLEU score, byte perplexity, and a custom verification function.

---

> ### Author Rebuttal · Authors · 2024-08-17
>
> **(1) While the dataset covers fundamental quantum algorithms, it may not fully represent the complexity of more advanced quantum algorithms used in research.**
>
> We appreciate the reviewer's suggestion and have implemented Generalized Simon's Problem to demonstrate the compatibility of our framework with more advanced quantum algorithms. We refer the reviewer to General Response (1) for more details. As we expand the QCircuitNet dataset, hopefully it will encompass quantum algorithms of different complexity levels and become useful for realistic quantum research.
>
>
> **(2) The binary nature of the verification function (correct/incorrect) may not capture partial correctness or near-misses in algorithm implementation.**
>
> We thank the reviewer for the comment. We would like to clarify that the scoring function is not binary. The details of verification function are provided in Section 4.2 and Appendix A.1. The verification function returns -1 for syntax errors and a score between 0 and 1 for syntactically correct functions. For the tasks that solve for a specific value, such as the secret string in Simon's problem and the marked item in Grover's algorithm, the score represents the success rate on test cases. For the tasks that prepare a specific state, such as generating the GHZ state or performing the quantum Fourier transform, the score is the fidelity of the generated state. This design was chosen to provide a more nuanced assessment beyond binary scoring. As demonstrated in the updated Table 1 \& 2 in the supplementary materials, verification scores range from -1 to 1, capturing the varying difficulty levels of different algorithms. From the perspective of algorithm design, it is challenging to automatically evaluate correctness without adopting accuracy-based verification standards. To better capture 'near-misses,' we also employ metrics like BLEU to measure the similarity between the model's output and the reference implementation. These machine learning metrics complement the verification function, providing a more comprehensive evaluation.
>
> **(3) The dataset focuses on circuit design and simulation but doesn't address the challenges of implementing these algorithms on actual quantum hardware with noise and errors.**
>
> We appreciate this valuable feedback. QCircuitNet is focused on quantum algorithm design from a theoretical perspective, thereby intentionally abstracting away from hardware-specific challenges such as noise and errors. We fully acknowledge that hardware-efficient implementation is a crucial research direction. Nevertheless, this topic involves highly specialized settings and techniques, such as qubit connectivity, basis gate set, error model, and so on. These go far beyond the scope of algorithm design, which is the main focus of our current study. We view this as a promising direction for future research.
>
> **(4) While the paper suggests the potential for fine-tuning LLMs, it doesn't provide concrete experiments or results in this direction.**
>
> We thank the reviewer for the suggestion. We initially left this as future work since we were targeting at a benchmark dataset and the unique nature of quantum data requires non-trivial fine-tuning techniques. We added experiments as suggested and refer the reviewer to General Response (4) for more details.
>
> **(5) While the paper presents performance metrics, it doesn't delve deeply into the types of errors made by LLMs or patterns in their mistakes.**
>
> We thank the reviewer for the comment. We would like to point out that in Section C.3 in Appendix, we included several case studies to illustrate and analyze various types of errors made by LLMs. For example, we provided concrete examples of mistakes made by GPT-4o in one-shot setting. Our analysis revealed that the lower scores in this setting might be due to GPT-4o’s tendency to improvise by drawing on pre-trained knowledge rather than closely following the syntax of the example, leading to avoidable "errors". This issue was significantly alleviated in the 5-shot setting, highlighting GPT-4o's strong in-context learning ability. We refer the reviewer to Appendix C.3 for more details.
>
> Another prominent type of error can be attributed to the limitations of LLMs in counting. In Oracle Construction tasks, such as those for the Bernstein-Vazirani problem, the model needs to count the positions of '1's in the secret string and apply gates to the corresponding qubits. This is a well-known challenge for LLMs, and our results once again confirmed this. Interestingly, as our new experiments showed, this issue might be partially mitigated through fine-tuning. For more details on the fine-tuning results, please refer to General Response (4).
>
> **(6) The paper doesn't thoroughly address how the approach scales with increasing qubit numbers or more complex quantum algorithms.**
>
> We thank the reviewer for the suggestion. In principle the framework of QCircuitNet is designed to scale easily with increasing qubit numbers. We refer the reviewer to General Response (3) for more detailed discussion.

---

> ### Author Response · Authors · 2024-08-26
>
> Dear Reviewer,
>
> Thank you once again for the time and effort you've devoted to reviewing our paper and considering our rebuttal. As we approach the end of the discussion period, we would greatly appreciate any further feedback you might have. Your insights are invaluable, and we are keen to address any additional concerns to improve our work.
>
> We look forward to your response.

---

### Author Rebuttal · Authors · 2024-08-17

# General Response

We thank all the reviewers for their efforts and valuable feedback. We noticed several common concerns and provide a general response as follows.

**(1) Call for more advanced quantum algorithms used in research.**

We appreciate the reviewers' suggestion. Extension to more advanced quantum algorithms was one of the core considerations when we designed the framework of QCircuitNet. To demonstrate how to apply the framework to more complex and advanced quantum algorithms, here we provide a concrete example by implementing Generalized Simon's Problem, a generalized and more advanced version of the standard Simon's problem and has been an active area of research in recent years [1, 2]. We implemented the quantum algorithm solving Generalized Simon's Problem from [1]. The setting is formally stated as follows: given an (unknown) function $f: \mathbb{Z}_p^n \rightarrow X$, where $X$ is a finite set, and a positive integer $k<n$, it is guaranteed that there exists a subgroup $S \leq \mathbb{Z}_p^n$ of rank $k$ such that for any $x, y \in \mathbb{Z}_p^n, f(x)=f(y)$ iff $x-y \in S$. The goal is to find $S$. (Intuitively, the generalized Simon's problem extends the standard Simon's problem from binary to $p$-ary bases and from a single secret string to a subgroup of rank $k$.)

We applied the same few-shot learning method to benchmark the performance of various LLMs on this advanced algorithm. The results are presented below.

*Table 1: Generalized Simon (multiple strings) Oracle Construction Scores*

| Model              | Setting      | BLEU             | PPL              | Verification        |
|--------------------|--------------|------------------|------------------|---------------------|
| gpt-4o-2024-05-13  | few-shot(5)  | 28.3031(±2.4775) | -                | -0.1875(±0.0701)     |
| Meta-Llama-3-8B    | few-shot(5)  | 20.7813(±1.8074) | 1.1563(±0.0041)  | -0.8125(±0.0701)     |

*Table 2: Generalized Simon (multiple strings) Algorithm Design Scores*

| Model              | Setting      | BLEU             | PPL              | Verification        |
|--------------------|--------------|------------------|------------------|---------------------|
| gpt-4o-2024-05-13  | few-shot(5)  | 26.2267(±1.1064) | -                | -1.0000(±0.0000)     |
| Meta-Llama-3-8B    | few-shot(5)  | 31.7937(±5.3745) | 1.4235(±0.0135)  | -1.0000(±0.0000)     |

*Table 3: Generalized Simon (3-ary base) Algorithm Design Scores*

| Model              | Setting      | BLEU             | PPL              | Verification        |
|--------------------|--------------|------------------|------------------|---------------------|
| gpt-4o-2024-05-13  | few-shot(5)  | 9.0448(±1.4988)  | -                | -1.0000(±0.0000)     |
| Meta-Llama-3-8B    | few-shot(5)  | 10.6289(±4.2092) | 1.3488(±0.0222)  | -1.0000(±0.0000)     |

We would like to assure the reviewers that QCircuitNet is a long-term project and will be actively updated and maintained in the future. With the efforts to integrate more advanced algorithms, we aim to expand the dataset's scale and range of difficulty levels, making it a valuable resource for research in quantum logic synthesis and quantum algorithms.

[1] Z. Ye, Y. Huang, L. Li, and Y. Wang. Query complexity of generalized simon’s problem. Information and Computation, 281:104790, 2021.

[2] Z. Wu, D. Qiu, J. Tan, H. Li, and G. Cai. Quantum and classical query complexities for generalized simon’s problem. Theoretical Computer Science, 924:171–186, 2022.

---

> ### Author Rebuttal · Authors · 2024-08-17
>
> **(2) Clarification of several experimental metrics.**
>
> We thank the reviewers for the detailed suggestions. We further clarified the scoring of verification functions in Response (2) to Reviewer 7rUP and provided the standard error for our experiment results in Response (3) to Reviewer DaGg.
>
> **(3) The scalability of the approach.**
>
> We appreciate the reviewers' insightful comments. In principle, our approach is designed to scale with increasing qubit numbers and more complex quantum algorithms. As emphasized by Reviewer DaGg, theoretically meaningful quantum algorithms which can be implemented efficiently should have no more than polynomial gates, therefore circuit formulation allows for scalable representations of quantum algorithms in theory. Regarding implementation details, the dataset generation scripts are written in a generic way which can be easily extended to arbitrarily large qubit numbers. We have also implemented Generalized Simon's Problem in General Response (1) which showcases the compatibility of our framework with more complex algorithms. The main bottleneck for scalability in the whole pipeline lies in the simulation process in verification function. Efficient classical simulation for non-Clifford quantum circuits is by nature difficult, which is also the motivation for developing quantum computers. However, the verification procedure does not need to rely solely on classical simulation; it can be complemented or replaced by real quantum hardware to create a hybrid workflow. As quantum hardware becomes more widely available and the capabilities of large language models continue to improve, the dataset can be easily expanded to accommodate this scaling. We believe that future advancements in these areas will allow our framework to better handle larger quantum systems and more complex algorithms.

---

> ### Author Rebuttal · Authors · 2024-08-17
>
> **(4) Training / Fine-tuning based on QCircuitNet dataset.**
>
> We thank the reviewers for the valuable suggestions. We also consider fine-tuning / training from scratch based on our dataset as an interesting and important research direction. This was initially left as future work since we were targeting at a benchmark dataset for now. Moreover, the unique nature of quantum data requires novel fine-tuning methods and model architecture designs, which presents significant research potential and could serve as a standalone topic. For a primitive demonstration, we conducted fine-tuning on data from the original Oracle Construction task and the newly generated random circuits, as suggested by Reviewer DaGg.
>
> Following [1], we quantize the model to 8-bits and then train it with LORA [2]. In our experiments, we use fp16 computational datatype. We set LORA $r=16,\alpha=32$ and add LORA modules on all the query and value layers. We also use AdamW [3] and LoRA dropout of $0.05$. The results are shown as follows:
>
> *Table 4: Supplementary Oracle Construction and Random Circuits BLEU Score*
>
> | Model              | Setting       | Bernstein-Vazirani    | Deutsch-Jozsa        | Grover               | Simon               | Clifford            | Universal           | Average |
> |--------------------|---------------|-----------------------|----------------------|----------------------|---------------------|---------------------|---------------------|---------|
> | gpt-4o-2024-05-13  | few-shot(5)   | 95.6388(±0.3062)      | 91.0564(±0.6650)     | 92.0620(±0.6288)     | 80.3390(±2.0900)    | 39.5469(±3.6983)    | 33.3673(±3.1007)    | 72.0017 |
> | Meta-Llama-3-8B    | few-shot(5)   | 53.5574(±5.2499)      | 69.8996(±5.7812)     | 61.3102(±5.4671)     | 26.3083(±2.0048)    | 13.0729(±0.9907)    | 13.4185(±1.2299)    | 39.5945 |
> | Meta-Llama-3-8B    | finetune      | 76.0480(±7.9255)      | 71.8378(±2.4179)     | 67.7892(±7.8900)     | 43.8469(±3.2998)    | 10.8978(±0.6169)    | 7.1854(±0.5009)     | 46.2675 |
>
> *Table 5: Supplementary Oracle Construction and Random Circuits Verification Score*
>
> | Model              | Setting       | Bernstein-Vazirani    | Deutsch-Jozsa        | Grover               | Simon               | Clifford            | Universal           | Average |
> |--------------------|---------------|-----------------------|----------------------|----------------------|---------------------|---------------------|---------------------|---------|
> | gpt-4o-2024-05-13  | few-shot(5)   | 0.0000(±0.0246)       | 0.4300(±0.0590)      | 0.0000(±0.1005)      | -0.0200(±0.0141)    | -0.0333(±0.0401)    | -0.1023(±0.0443)    | 0.0457  |
> | Meta-Llama-3-8B    | few-shot(5)   | -0.2700(±0.0468)      | 0.0900(±0.0668)      | -0.5200(±0.0858)     | -0.6600(±0.0476)    | -0.7303(±0.0473)    | -0.5056(±0.0549)    | -0.4327 |
> | Meta-Llama-3-8B    | finetune      | -0.1300(±0.0485)      | -0.2000(±0.0402)     | -0.3300(±0.0900)     | -0.7400(±0.0441)    | -0.8741(±0.0343)    | -0.9342(±0.0262)    | -0.5347 |
>
> *Table 6: Supplementary Oracle Construction and Random Circuits Perplexity Score*
>
> | Model              | Setting      | Bernstein-Vazirani | Deutsch-Jozsa     | Grover             | Simon              | Clifford          | Universal         | Average  |
> |--------------------|--------------|--------------------|-------------------|--------------------|--------------------|-------------------|-------------------|----------|
> | Meta-Llama-3-8B    | few-shot(5)  | 1.1967(±0.0028)     | 1.1174(±0.0015)   | 1.1527(±0.0021)    | 1.1119(±0.0017)    | 1.4486(±0.0054)    | 1.4975(±0.0051)    | 1.2541   |
> | Meta-Llama-3-8B    | finetune     | 1.0004(±0.0002)     | 1.1090(±0.0014)   | 1.0010(±0.0006)    | 1.1072(±0.0011)    | 1.2944(±0.0053)    | 1.3299(±0.0055)    | 1.1403   |
>
> We compared the performance of Llama3-8B before and after fine-tuning with case studies. Take oracle construction of Bernstein-Vazirani Problem as an example, we observed that before fine-tuning, the model would indiscriminately apply CX gates to all qubits. After fine-tuning, it began to selectively apply CX gates to qubits with '1's in the secret string. In some cases, the positions were still counted incorrectly; however, in certain instances, the model accurately identified all the positions for applying the CX gates, which is impressive from our perspective. This improvement significantly contributed to higher scores, suggesting that the model is starting to learn the pattern for constructing certain oracles through fine-tuning.
>
> [1] T. Dettmers, A. Pagnoni, A. Holtzman, and L. Zettlemoyer. Qlora: Efficient finetuning of quantized llms. Advances in Neural Information Processing Systems, 36, 2024.
>
> [2] E. J. Hu, yelong shen, P. Wallis, Z. Allen-Zhu, Y. Li, S. Wang, L. Wang, and W. Chen. LoRA: Low-rank adaptation of large language models. In International Conference on Learning Representations, 2022.
>
> [3] I. Loshchilov and F. Hutter. Decoupled weight decay regularization. In International Conference on Learning Representations, 2019.

---

### Author Response · Authors · 2024-08-22

Dear Reviewers,

We would like to express our sincere gratitude for the time and effort you have dedicated to reviewing our paper. We have carefully considered your feedback and tried our best to provide a detailed and thorough rebuttal in response.

As the discussion period approaches its halfway point, we are eager to hear your opinions and to address any further concerns you may have. Your insights are crucial to the improvement of our work, and we are committed to engaging in a constructive dialogue.

Thank you once again for your dedication to this process, and we look forward to your response.

---

### Decision · Program_Chairs · 2024-09-26

**Decision:**

Reject

**Comment:**

The paper introduces QCircuitNet, a large-scale hierarchical dataset designed for quantum algorithm design, leveraging LLMs to handle the challenging task of generating quantum circuits. This work is potentially significant in bridging the gap between quantum computing and AI by proposing a dataset that could serve as a valuable benchmark for AI's capabilities in designing quantum algorithms. The authors provide an innovative framework for formulating quantum algorithm design as a language modeling task, and the dataset is thoughtfully structured, encompassing critical quantum primitives, problem descriptions, and validation functions.

However, the paper falls short in several key areas, warranting a rejection based on the high standards of NeurIPS D&B track.

First, while the dataset introduces fundamental quantum algorithms, the scope is limited to toy problems, and the lack of inclusion of advanced quantum algorithms weakens its relevance to cutting-edge research. Furthermore, the authors do not fully address the scalability of their approach when it comes to increasing the number of qubits or the complexity of quantum algorithms. The results of using LLMs on these tasks are not promising, as evidenced by poor BLEU scores and perplexity values, and the paper lacks a detailed exploration of the types of errors the LLMs make. Additionally, the experimental results are mostly limited to few-shot learning, with little emphasis on the potential of fine-tuning the models, making the experimental depth insufficient.